# Whole-genome microsynteny-based phylogeny of angiosperms

Tao Zhao [1,2,3 ✉], Arthur Zwaenepoel[2,3], Jia-Yu Xue[4,5], Shu-Min Kao[2,3], Zhen Li[2,3], M. Eric Schranz [6] & Yves Van de Peer [2,3,4,7 ✉]

Plant genomes vary greatly in size, organization, and architecture. Such structural differences may be highly relevant for inference of genome evolution dynamics and phylogeny. Indeed, microsynteny—the conservation of local gene content and order—is recognized as a valuable source of phylogenetic information, but its use for the inference of large phylogenies has been limited. Here, by combining synteny network analysis, matrix representation, and maximum likelihood phylogenetic inference, we provide a way to reconstruct phylogenies based on microsynteny information. Both simulations and use of empirical data sets show our method to be accurate, consistent, and widely applicable. As an example, we focus on the analysis of a large-scale whole-genome data set for angiosperms, including more than 120 available high-quality genomes, representing more than 50 different plant families and 30 orders. Our 'microsynteny-based' tree is largely congruent with phylogenies proposed based on more traditional sequence alignment-based methods and current phylogenetic classifications but differs for some long-contested and controversial relationships. For instance, our synteny-based tree finds Vitales as early diverging eudicots, Saxifragales within superasterids, and magnoliids as sister to monocots. We discuss how synteny-based phylogenetic inference can complement traditional methods and could provide additional insights into some long-standing controversial phylogenetic relationships.

---

[1] State Key Laboratory of Crop Stress Biology for Arid Areas/Shaanxi Key Laboratory of Apple, College of Horticulture, Northwest A&F University, Yangling, China. [2] Department of Plant Biotechnology and Bioinformatics, Ghent University, Ghent, Belgium. [3] Center for Plant Systems Biology, VIB, Ghent, Belgium. [4] College of Horticulture, Academy for Advanced Interdisciplinary Studies, Nanjing Agricultural University, Nanjing, China. [5] Institute of Botany, Jiangsu Province and Chinese Academy of Sciences, Nanjing, China. [6] Biosystematics Group, Wageningen University and Research, Wageningen, The Netherlands. [7] Center for Microbial Ecology and Genomics, Department of Biochemistry, Genetics and Microbiology, University of Pretoria, Pretoria, South Africa. ✉email: tao.zhao@nwafu.edu.cn; yves.vandepeer@psb.ugent.be

**M**icrosynteny (hereafter also simply referred to as synteny), or the local conservation of gene order and content, provides a valuable means to infer the shared ancestry of groups of genes and is commonly used to infer the occurrence of ancient polyploidy events[1,2], to identify genomic rearrangements[3], and to establish gene orthology relationships[4–6], particularly for large gene families where sequence-based phylogenetic methods may be inconclusive[7,8]. In addition, microsynteny has also been used for the inference of phylogenetic relationships. However, phylogenetic inference using synteny information presents notorious algorithmic and statistical challenges and tends to be computationally costly[9–12]. As a result, studies on synteny-based phylogenetic inference have been rare and mostly restricted to relatively simple data sets such as plastid[13] or bacterial genomes[14,15]. Recently, Drillon et al[16]. developed a distance-based method based on breakpoints of synteny blocks. This method was successfully applied to 13 vertebrate genomes and 21 yeast genomes[16]. For likelihood-based approaches, the framework used in Lin et al[17]. and its variants[18–20] showed potential for handling bigger data sets[17]. However, such methods convert absolute two-gene adjacencies into binary sequences as a proxy of genomic structure[17,20], which does not allow considering variable degrees of proximity created by fractionation[21].

Plant genomes are highly diverse in size and structure[22,23] and sculpted by a complex interplay of small- and large-scale duplication events[24,25] and subsequent fractionation, hybridization[26], transposable element (TE) activity[27], and gene loss[28,29]. Consequently, employing synteny information for phylogenetic inference from large plant genome datasets has been notoriously difficult. Recently, some of us developed an approach in which microsynteny information is converted into a network data structure, and which has proven to be well suited for evolutionary synteny comparisons among many eukaryotic nuclear genomes[7,23]. Using this approach, conservation or divergence of genome structure can be conveniently summarized and reflected by synteny cluster sizes and composition. Importantly, the network representation of synteny relationships provides an abstraction of structural homology across genomes that is in principle amenable to tree inference using standard approaches from phylogenetics, however this possibility has hitherto not been explored.

Here, we combine synteny networks and their matrix representation with standard maximum-likelihood based statistical phylogenetics to reconstruct phylogenies for large whole-genome data sets. After validation of our approach using simulated and empirical whole-genome data sets, we construct a well-resolved 'synteny tree' using currently available representative genomes of flowering plants. The obtained tree is highly congruent with current phylogenetic classifications, although some notable differences concerning the phylogenetic positions of magnoliids, Vitales, Caryophyllales, Saxifragales, and Santalales were observed. We argue that our approach provides a noteworthy complement to more classical approaches of tree inference and could help to solve some longstanding problems that may remain difficult to solve with sequence (alignment) based methods. Moreover, we believe the synteny network data structure we advocate here may provide a promising avenue for further research into model-based statistical phylogenetic inference from whole-genome data.

## Results

**'Synteny matrix representation with likelihood', or Syn-MRL**. Our synteny-based phylogenetic reconstruction approach proceeds by encoding phylogenomic synteny pattern profiles, obtained from a clustering of the synteny network, into a binary data matrix (Fig. 1). We analyze this data matrix using a standard maximum likelihood (ML) reconstruction program under a binary model (here we use Mk) (Fig. 1). Such a process is similar to the MRL (Matrix Representation with Likelihood) supertree method (which uses the same data matrix as Matrix Representation with Parsimony, but with ML-based inference, yielding higher accuracy[30,31]), except that MRL is based on a set of input trees, in contrast with our synteny matrix, which is based on synteny clusters. To avoid misunderstanding, in this study we confine the usage of 'MRL' to the supertree method; and we refer to our 'supercluster' approach as Syn-MRL (Synteny Matrix Representation with Likelihood).

**Simulated data**. We conducted simulation studies to evaluate the statistical performance of the proposed approach (Fig. 2). To this end, we specified a continuous-time Markov model for the evolution of gene families along a species tree with syntenic edges between genes and simulated 'gene family syntenic networks' in stages across a given dated species tree using a Gillespie-like algorithm (see Methods) (Fig. 2). The resulting syntenic networks were analyzed using the Syn-MRL approach and we assessed the topological accuracy of the phylogenies inferred from the simulated data by computing the Robinson–Foulds (RF) distance[32] to the true tree.

We first aimed to verify the accuracy and statistical consistency of the approach under the assumed model of evolution. Our simulations were based on a 15-monocot species tree. To ensure our simulations employ reasonable parameter values, we first estimated appropriate gene duplication ($\lambda$) and loss rates ($\mu$) based on the corresponding gene content matrix (see Methods). Based on these estimates, we use a single gene turnover rate (i.e. assume $\lambda = \mu$) of 0.38 events per gene per 100 M$y$. Values for $p_r$ and $p_d$ were sampled independently from a Beta(b,b) distribution, where b was drawn from a Beta(2,2) distribution for each simulation replicate. We further use a Geometric prior with parameter $p = 0.66$ on the number of genes at the root of the species tree in each family. We simulated data sets of different sizes, consisting of 1000, 2000, 5000, 10,000, 20,000, and 50,000 families, with 1000 replicate simulations for each data set size. Results for these simulations showed that increasing the total number of gene families (and, concomitantly, synteny clusters) leads to consistently better inferences (increasing proportion of RF = 0 trees) (Fig. 2b).

Next, we used a 62-plant species tree (derived from the time tree reported by Morris et al.[33]) as input tree to examine the accuracy of the recovered species tree as a function of gene duplication and loss rates ($\lambda$ and $\mu$, note that we still assume $\lambda = \mu$), and rearrangement rate ($\nu$). We generated simulated data sets of 1000 gene families with 1000 replicates (for the sake of computational tractability). The result showed that even with many more input species (which allows for a bigger variance of the inferred RF distances) and a relatively small number of gene families being analyzed, the tree topology with the highest frequency is still the correct tree (Fig. 2c). Tree accuracy decreases with increasing duplication and loss rate, especially when the rate is higher than 1 (Supplementary Fig. 1a). However, this is much less so when $\lambda$ is less than 0.5 (Supplementary Fig. 1a). A similar pattern is observed for the relationship of the rearrangement rate and the RF distance of the recovered tree, but with a much lower correlation (Supplementary Fig. 1b). These results indicate that a higher rate (e.g. between 0.5 and 1 in our simulations) of $\nu$ does not necessarily entail lower accuracy, as it may also result in useful phylogenetic signals (Supplementary Fig. 1b).

**Tests on empirical genome data sets**. We further evaluated the accuracy of our approach on data sets that have been studied in synteny-based phylogenetic reconstructions: 19 yeast genomes

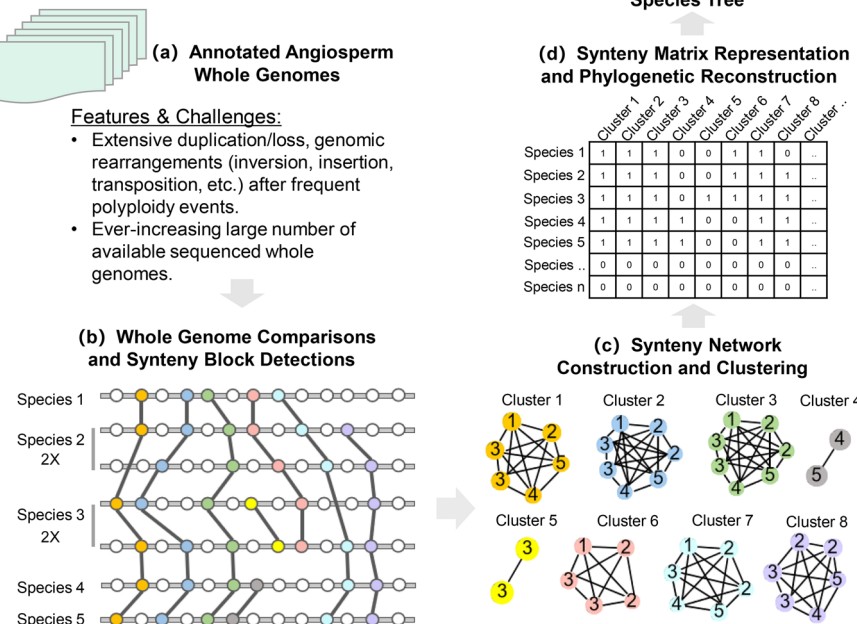

**Fig. 1 Whole-genome microsynteny-based species tree inference. a** Whole-genome data sets with all predicted genes are used for phylogeny reconstruction. **b** The synteny network approach first conducts all pairwise reciprocal genome comparisons, followed by synteny block detection. Syntenic anchor pairs from all syntenic blocks constitute the synteny network database (see Methods for details). **c** We analyzed all synteny clusters after clustering the entire network database. Synteny clusters vary in size and node compositions. Shared genomic rearrangements are reflected by cluster compositions. Specific anchor pairs shared by a lineage/species form specific clusters (e.g. Clusters 4–6). We account for the presence or absence of the same recurring anchors for multiple blocks derived from whole-genome or segmental duplications (e.g. for Species 2 and 3 in Clusters 2, 3, 5, and 8). **d** The phylogenomic profiling of all clusters constructs a binary matrix, where rows represent species and columns represent clusters. The synteny matrix comprehensively represents phylogenomic gene order dynamics. It transforms the concept of synteny comparisons from analyzing massive parallel coordinates plots into analyzing profiles of individual clusters/networks. Each cluster stands for a shared homologous 'context'. For example, TE activity can cause genes to be transposed as insertions into new contexts or be lost from the original context (e.g. genes in Clusters 4–6). As long as such transpositions are shared by different genomes (e.g. genes in Clusters 4 and 6) or within the same genome because of whole-genome duplication (e.g. genes in Cluster 5), specific clusters will emerge and corresponding signals will be added to the matrix. This synteny matrix is used as the input for species tree inference by maximum likelihood (referred to as Syn-MRL).

used in YGOB (Yeast Gene Order Browser)[34], 21 yeast genomes, as used in another study[16], 12 *Drosophila* genomes[35], and 13 vertebrate genomes[16]. We refer to the two yeast data sets as the 'YGOB' and 'yeast' data, respectively. For each data set, we constructed synteny networks and encoded phylogenomic profiles of all synteny clusters into a binary matrix (synteny matrix), which we used to infer a phylogenetic tree with the MRL approach.

We downloaded genome annotations for each set of the species directly from the public domain (Supplementary Data 1). Note that for some genomes, required genome annotations used in Syn-MRL may not be accessible, in such cases we used available genomes within the same genera as substitutes (Supplementary Data 1). The genome data sets varied in genome size, contiguity, and evolutionary distances, all of which can have an impact on synteny detection. We investigated how synteny conservation and parameter settings can have impacts to the final inferred tree on different data sets. To this end, we ran Syn-MRL under different settings of 'minimum number of required anchors' ($A_{\min}$, where anchor refers to a pair of homologous genes with conserved neighborhoods) for synteny block calling. We compared synteny network metrics[23], including pairwise syntenic percentage of each genome comparison (number of syntenic genes relative to the total number of genes between two genomes), network status including number of nodes and edges, connectivity (clustering coefficient and node degree), total number of clusters, and cluster composition (number of involved species per cluster) (Supplementary Figs. 2–5) under seven settings of $A_{\min}$ ('-s' setting of MCScanX[36]) (at this step we use a setting of 25 maximum gene

gaps ($G_{\max} = 25$)). Notably, for the tested vertebrate and yeast data sets, stricter parameters led to a substantial increase of smaller syntenic percentages (Supplementary Figs. 2 and 3d). For network status, increasing $A_{\min}$ generally leads to fewer nodes and edges, and lower node degree, but concomitantly results in more scattered clusters (Supplementary Figs. 4 and 5).

We compared the robustness (indicated by bootstrap support values) and topological differences of the inferred trees under different settings, and compared these with the sequence-alignment (SA) based trees and the reported trees in the associated studies (Fig. 3a–e). In general, our method constructed highly comparable species trees for the tested data sets under more permissive and moderate synteny parameter settings ($A_2 G_{25}$, $A_3 G_{25}$, and $A_5 G_{25}$, where we write $A_x G_y$ as a shorthand for $A_{min} = x$, $G_{max} = y$). Increased topological uncertainties were observed when $A_{\min}$ exceeds 7 (Fig. 3a–e, Supplementary Figs. 6–9). Depending on the data set, stricter synteny parameter settings can lead to a substantial decrease, or even a total lost of nodes for certain species. This was observed for both the fish group in the vertebrate data set, and the yeast data set (where two divergent groups share little synteny, and the species '*Yarrowia lipolytica*' is absent from the network at $A_{11}$ and $A_{13}$) (Supplementary Fig. 2). Thus, a lack of data can lead to unreliable and lowly supported reconstructed topologies relating these genomes and clades (Supplementary Figs. 8 and 9). Comparatively, for the data sets of YGOB and *Drosophila*, a decent degree of synteny conservation was retained for most of the pairwise genome comparisons even at very strict parameters (Supplementary Fig. 2). Accordingly, the species trees reconstructed

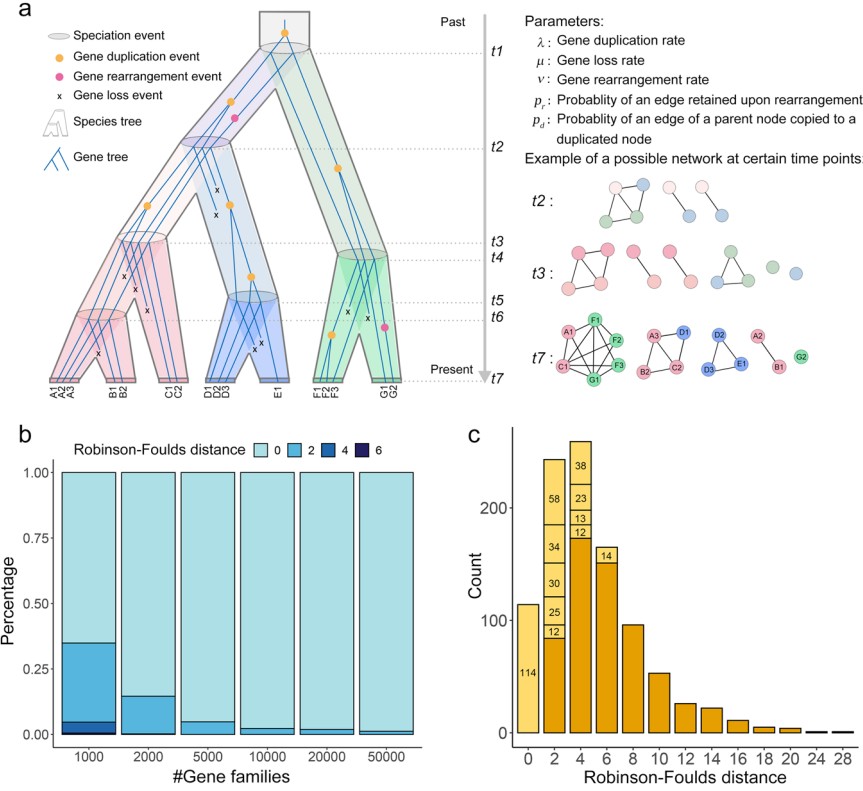

**Fig. 2 Results of Syn-MRL on simulations. a** Simulation process. An evolutionary scenario for a single gene family is illustrated, involving gene duplication, loss, and rearrangement. Extant gene names are labeled at the bottom (left panel). Nodes and edges of synteny clusters are simulated in stages under the listed parameters, 'snapshots' of the evolution of the associated gene family synteny network are shown at three-time points right after a speciation event (right panel). For example, at $t2$, three connected components can be formed: a 'conserved' four-node cluster, and two 'specific' two-node clusters, which were formed as a result of the gene duplication or rearrangement events that happened between $t1$ and $t2$. **b** Proportions of inferred trees with a particular RF distance to the true 15 monocot species tree. Simulations were conducted for different numbers of total gene families, with 1000 replicates for each simulation. **c** Distribution of RF distances of the inferred trees to the true 62 plant species tree. A total of 1000 simulation replicates of 1000 gene families were conducted. The 10 most frequently recovered topologies are labeled within the corresponding bar of the histogram.

are highly consistent and well-supported for these two data sets (Supplementary Figs. 6 and 7). The inferred phylogeny for the *Drosophila* data set does not vary and receives high overall bootstrap support across settings (Supplementary Fig. 7).

Interestingly, for the vertebrate data set, the monophyly of birds was recovered at $A_{min} = 9$, 11, and 13 (Supplementary Fig. 8e–g). However, under the same settings topological uncertainties emerged for the three fish genomes (Supplementary Fig. 8). We visualized the changes of binary matrices used for the ML estimation for this data set under different settings (Supplementary Fig. 10) and consider the *Drosophila* data set as a comparison (from which the reconstructed trees were consistent) (Supplementary Fig. 11). The proportion of fish-genomes derived signals (columns) is smaller and less dynamic, and bird-specific signals become more pronounced at stricter settings (Supplementary Fig. 10). Analysis based on node percentages (the proportion of nodes for a specific taxon in the syntenic network) also revealed that the total number of bird nodes are underrepresented in the total nodes of networks at more permissive settings (e.g. $A_2$ and $A_3$), whereas the total number of fish nodes are underrepresented at stricter settings (e.g. $A_7$ onwards) (Supplementary Fig. 12).

Next we adjusted the number of maximum gene gaps allowed ($G_{max}$) and explored more parameter settings for the vertebrate data set (Supplementary Fig. 13). As before, among the parameters being tested, we found that when the recovered

phylogeny includes the proper grouping of the fish genomes, birds tend not to be recovered as monophyletic and vice versa (Supplementary Fig. 13), the former being associated with rather liberal parameter settings while the latter with stricter settings ($A_3G_{(3,4,5)}$ and $A_5G_{(6,7)}$ compared to $A_3G_{(1,2)}$ and $A_5G_{(4,5)}$, respectively) (Supplementary Fig. 13). However, the bird group and the fish group in the vertebrate data set can both be resolved as expected at '$A_3G_2$' and '$A_5G_6$' (both with lower bootstrap support values though) (Supplemental Figs. 13b and 13h).

The topological differences between the phylogenetic trees we reconstructed and the reported ones reflect some long-term well acknowledged phylogenetic controversies that remain unsettled to date (Figs. 3f–i). A detailed discussion of these discrepancies is beyond the scope of our study, and we refer to previous studies that have discussed these phylogenetic incongruencies. Examples include the position of *Candida glabrata*[37–39](Fig. 3f), the relationship of *Drosophila yakuba* and *D. erecta*[35,40](Fig. 3g), the relationships of Primates, Rodentia, and Laurasiatheria (the so called primate–rodent–carnivore controversy)[41–44] (Fig. 3h), and the relationships of *Kluyveromyces lactis* and *Eremothecium gossypii*[38,45–47](Fig. 3i).

Overall, our approach reconstructed highly comparable and resolved phylogenies for the four tested empirical data sets. Nevertheless, depending on the data set (taxon sampling, phylogenetic distances, and synteny properties), multiple parameter settings should be considered for pairwise synteny

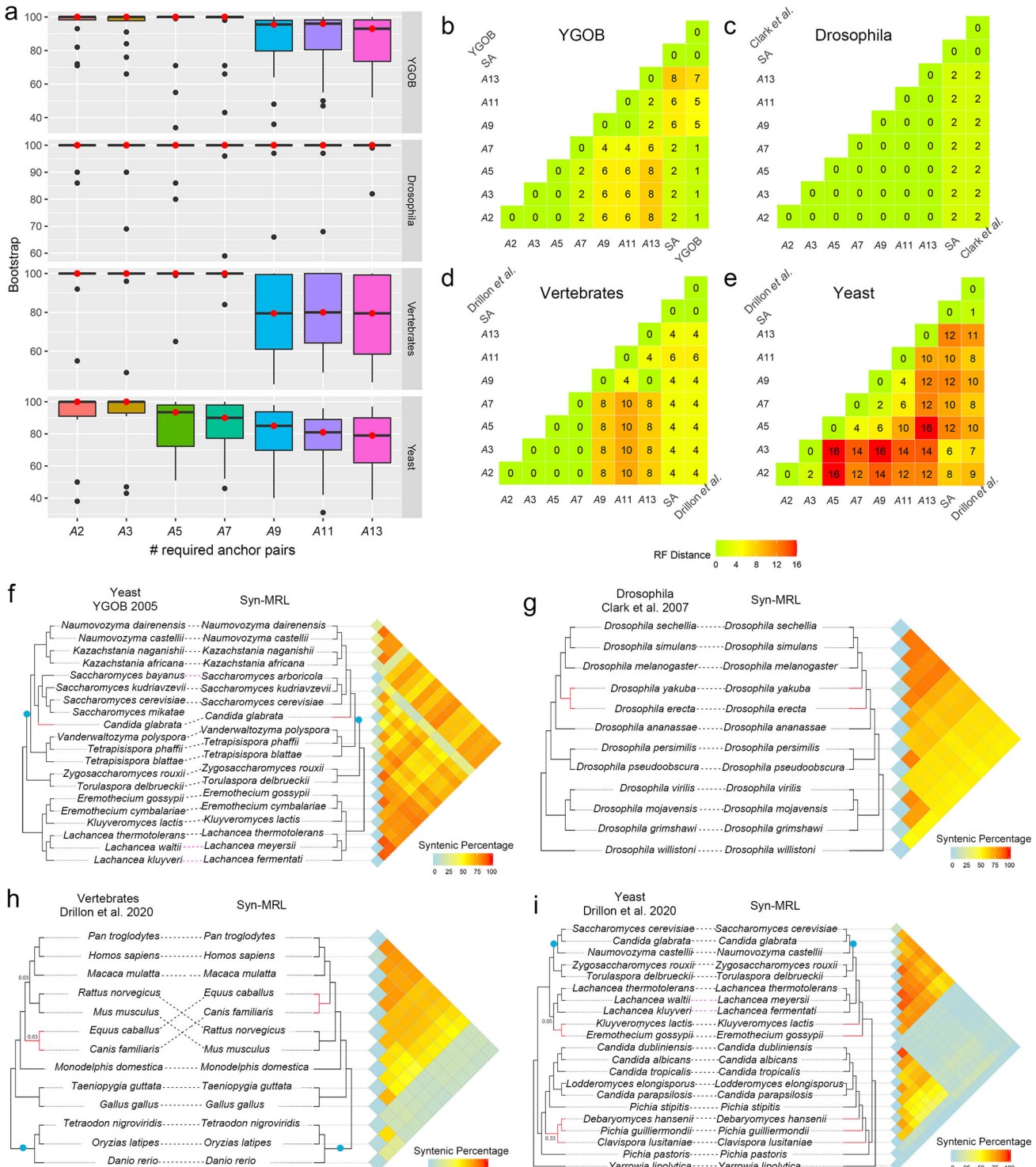

**Fig. 3 Results of Syn-MRL on empirical data sets. a** Distribution of the support values of all the nodes of the inferred trees for the four data sets under different settings of $A_{min}$ (the number of required anchor pairs for regions to be called syntenic). The boxplots indicate the minimum, maximum, median (the middle hinge with red dots), first quartile (the lower hinge), and third quartile (the upper hinge) in the data sets ($n = 16$, 9, 10, and 18 for the data sets of YGOB, *Drosophila*, vertebrates, and yeast). The whiskers represent the 1.5 inter-quartile range (IQR) extending from the hinges and outliers are shown as individual points. **b–e** Pairwise RF distances of the inferred trees, sequence alignment-based tree, and the reported tree of the data sets of YGOB, *Drosophila*, vertebrates, and yeast, respectively. Note that for the yeast data set, the species 'Yarrowia lipolytica' (used as the root of the tree) was absent from the data matrices at $A_{11}$ and $A_{13}$ (no synteny detected), and we therefore manually added this species (as the root) to the inferred tree, in order to calculate the RF distances. **f–i** Tree comparisons between the reported tree and the Syn-MRL tree of YGOB, *Drosophila*, vertebrates, and yeast data sets, at parameter setting values of $A_5G_{25}$, $A_5G_{25}$, $A_5G_6$, and $A_2G_{25}$, respectively. Matrices of syntenic percentages of pairwise genome comparisons are aligned to the corresponding species. Each cell of the matrix represents an overall syntenic percentage of a genome comparison, which is calculated using the total number of syntenic genes relative to the total number of genes between two genomes. The color indicates the values and goes from low (blue) to high (red). Different branching patterns are highlighted in red, genome duplication events are labeled as blue dots. Branch lengths are not meaningful. Support values for certain nodes of the vertebrate and yeast trees are from Drillon et al.[16].

comparisons. For data sets consisting of genomes with distinct genomic structures and properties, relying on a single parameter setting might be problematic to best represent synteny dynamics.

**Highly-resolved microsynteny-based phylogeny for angiosperms.** Our main aim was to infer a whole-genome synteny-based phylogeny for a large set of angiosperms. In angiosperms, the evolution of genome structure is particularly dynamic due to recurrent polyploidization—rediploidization cycles and massive TE activity. Furthermore, for several clades, phylogenetic relationships are often

contentious. After quality control, 123 fully sequenced plant genomes were used for synteny-based phylogenetic analysis (Supplementary Data 2), which includes synteny network construction and clustering, matrix representation of synteny followed by maximum likelihood estimation (see Methods for details). We used the parameter setting of '$A_5G_{25}$' for synteny detection, which has been tested and found to be most appropriate for the analysis of angiosperm genomes[23]. The size of the matrix used for ML tree inference is $123 \times 137,833$, which contains a binary presence/absence coding for each cluster in the synteny network (Fig. 1). The resulting best ML tree demonstrated

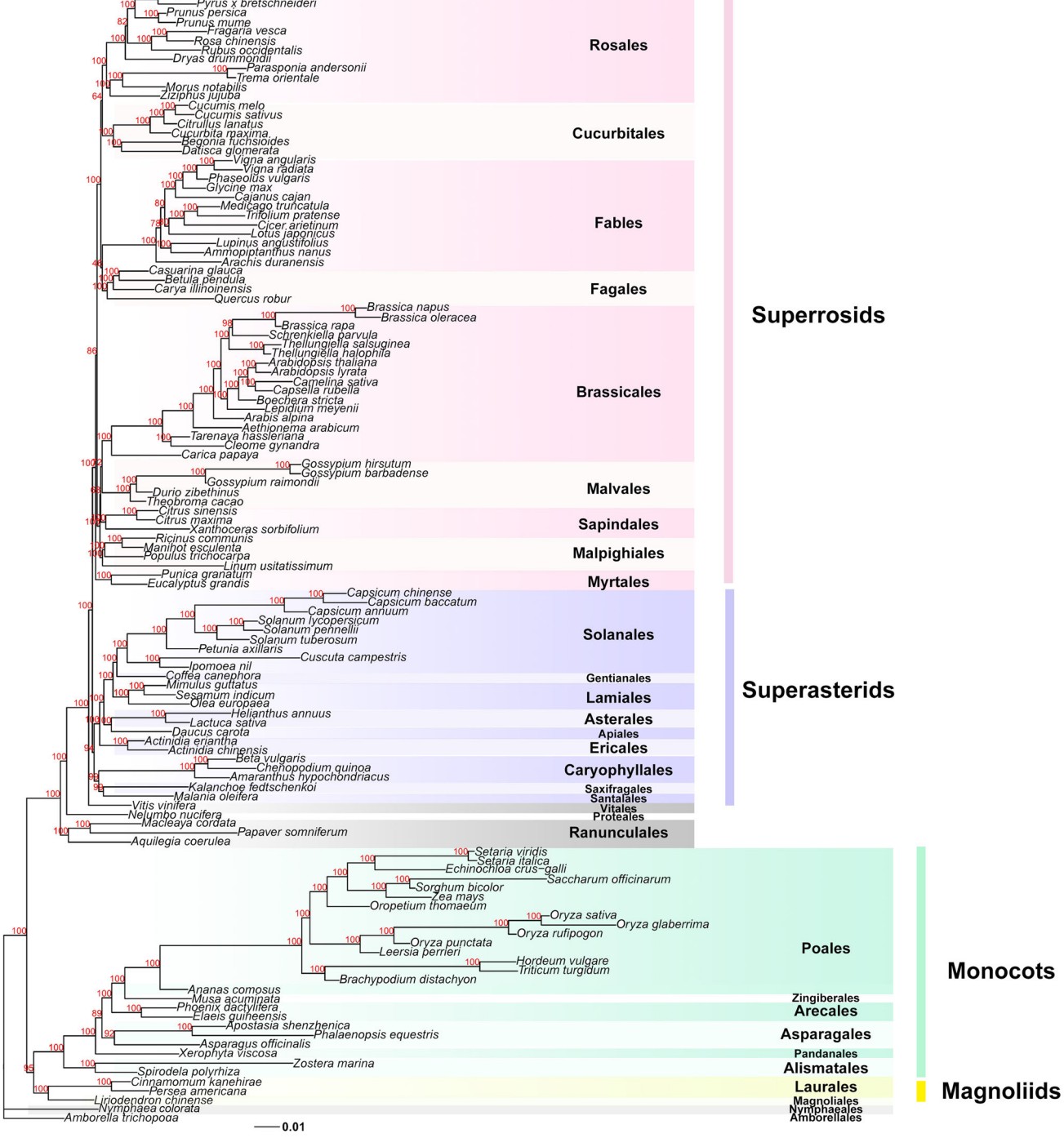

**Fig. 4 Maximum likelihood (ML) tree for 123 fully sequenced flowering plant genomes based on the microsynteny approach.** The tree is rooted by *Amborella*, and four main clades, i.e. superrosids, superasterids, monocots, and magnoliids are shaded in light-red, light-purple, light-green, and light-yellow, respectively. Ultrafast bootstrapping values are denoted for all the nodes. Names for the different plant orders follow the APG IV classification[48].

that the overall monophyly of most clades was strongly supported, and 110 of the 122 nodes had ≥95% bootstrap support (Fig. 4). *Amborella* was used as the sister group to other angiosperms in order to root the obtained phylogenies. Nymphaeales was resolved as a successive lineage right after *Amborella*, again sister to all other angiosperm genomes (Fig. 4). The monophyly of mesangiosperms was also strongly supported (Fig. 4). Interestingly, magnoliids (including Laurales and Magnoliales) form a sister group to monocots (BS = 95%), with the resulting clade sister to the eudicots (BS = 100%). Notably, our synteny tree strongly favors Vitales as sister to the rest of the core eudicots, branching off after Proteales (*Nelumbo*) (both BS = 100%) (Fig. 4). Moreover, in the synteny tree, we find Santalales as sister to Saxifragales (BS = 99%), and with both sister to Caryophyllales (BS = 99%), which in turn was found as sister to all other asterids (BS = 94%). Synteny trees positioned Myrtales (*Eucalyptus* and *Punica*) as early diverging rosids (BS = 100%), and Malpighiales as early diverging Malvids (BS = 100%). Almost all the nodes within Brassicales were fully resolved (Fig. 4). The monophyly of the nitrogen-fixing clade was fully recovered, and supports a relationship of ((Fagales, Fabales), (Rosales, Cucurbitales)), however with lower support (46% and 64% BS support, respectively) (Fig. 4).

### Comparison of the synteny tree with sequence alignment-based phylogenies and with current phylogenetic classifications.
Taking advantage of the large whole-genome plant data set used in this study, we also reconstructed phylogenies using widely adopted sequence alignment-based approaches for comparison with our microsynteny-based tree. We used the BUSCO gene set (v3.0), which is widely adopted in benchmarking genome assembly and annotation quality. Besides BUSCO, we developed a set of CSSC (Conserved Single-copy Synteny Clusters) gene markers from a profiling and screening of all synteny clusters (see Methods). We used supermatrix, supertree, and multispecies coalescent methods as representative sequence-based approaches (Supplementary Fig. 14) for these two sets of gene markers and inferred six phylogenetic trees (Supplementary Figs. 15–20). Overall, the six sequence alignment-based trees (hereafter referred to as SA trees) are highly similar, and only differ by a few bootstrap support values and a few minor sister-group relationships (Supplementary Figs. 15–20). We used the supermatrix-BUSCO tree as the representative of the SA trees and compared it with the synteny tree.

Comparing both trees shows that the synteny tree we have obtained is highly congruent with the SA tree (Supplementary Fig. 16). Interestingly, some notable differences were found between the two trees, such as the positioning of magnoliids, Santalales, Zingiberales, and Gentianales (Supplementary Fig. 16). Additional subtle differences were found within the orders, such as in Poales and Brassicales (Supplementary Fig. 16). We further compared our synteny tree with the current Angiosperm Phylogeny Group (APG) phylogeny (version IV)[48], and the 1000 plant transcriptomes (1KP) phylogeny of green plants[25] at the order level (Supplementary Figs. 22–24). Apart from the orders that do not have representative genomes yet, our synteny tree generally shows strong congruence to both the APG and 1KP trees (note however that a certain degree of incongruence exists between the two latter trees (Supplementary Fig. 22)) (Supplementary Figs. 23, 24). Besides the relationship of magnoliids and monocots and dicots, all discrepancies are confined to the positions and relationships of Vitales, Santalales, Saxifragales, Caryophyllales (Supplementary Figs. 22–24), which reflect long-time controversies in plant systematics and have been well acknowledged in the literature[25,48–56]. Our SA trees agree with the 1KP tree on magnoliids being sister to eudicots, which is different from the synteny tree (magnoliids sister to monocots) and APG tree (magnoliids sister to both monocots and eudicots) (Supplementary Figs. 21–24); the APG tree and the 1KP tree resolve Vitales and

Saxifragales as early diverging lineages within Rosids, whereas our SA trees (except the ASTRAL-BUSCO tree) and synteny tree favor Vitales as sister to the rest of the core eudicots, and Saxifragales cluster within Superasterids (Supplementary Figs. 15-24). We tested whether our synteny matrix (under the current synteny network construction parameters) would reject representative alternative topologies. To this end, we used the 'approximately unbiased' (AU) test[57] to evaluate the support for alternative topologies regarding the positions of magnoliids, Vitales, Saxifragales, Caryophyllales, and Myrtales (Supplementary Fig. 25). The test shows that our ML synteny tree was found to be significantly better than the alternative topologies with all alternative topologies rejected at the $p = 0.05$ level, except for the scenario where magnoliids are sister to eudicots ($p = 0.146$, Supplementary Fig. 25a). This means that at least based on our synteny matrix and the (admittedly ad hoc) Markov model of character evolution, alternative scenarios such as magnoliids sister to both monocots and eudicots (Supplementary Fig. 25b), Vitales (Supplementary Fig. 25c), or Saxifragales (Supplementary Fig. 25d) as early diverging rosids, and so on, were significantly less well supported.

### Insights towards ancestral introgressive hybridization.
Within the same plant family or order, our synteny tree also revealed some subtle differences with the SA trees (Supplementary Fig. 21). For example, the position of *Boechera* and *Arabis* in Brassicaceae, and the position of the Pooideae clade (including wheat, barley, and *Brachypodium*) in Poales (Supplementary Fig. 21).

Interestingly, a recent study has comprehensively investigated the particular branching order of three monophyletic groups in the Brassicaceae: *Boechera* (Clade B), *Capsella* and *Camelina* (Clade C), and *Arabidopsis* (Clade A)[58]. Introgression can be statistically tested by comparing the divergence times of the nodes (also referred as node depth or node height) from the trees supporting conflicting phylogenetic relationships[58,59]. Forsythe et al. proved the existence of massive nuclear introgression between Clades B and C, leading to substantially reduced sequence divergence between these clades[58]. As a result, the majority of single-copy gene trees strongly support the branching order of (A,(B,C)), while the true branching order is supposed to be (B,(A,C))[58]. Interestingly, our results are congruent with the findings of Forsythe et al.[58]. Alignment-based trees (supermatrix, MRL, and ASTRAL) all support (A,(B,C)) (Supplementary Figs. 15–20), whereas only the synteny tree supports the branching order (B,(A,C)) proposed by Forsythe et al.[58].

We used representative genomes of Clade A (*A. lyrata* and *A. thaliana*), Clade B (*B. stricta*), and Clade C (*C. rubella* and *C. sativa*), as well as an outgroup (*A. arabicum*) (we refer to this dataset as the ABC data set) to further investigate the reason of the divergence observed based on gene sequences and synteny. We identified low-copy gene orthogroups that were retained in all species (see Methods) followed by phylogenetic tree reconstruction for each orthogroup. In parallel we also performed Syn-MRL. Of the 6,306 trees, 3,713 (58.9%) supported the topology of (A,(B, C)) (BC topology), 1,164 (18.5%) supported (B,(A,C)) (AC topology) and 1,429 (22.6%) supported (C,(A,B)) (AB topology) (Supplementary Fig. 26a, Supplementary Data 3). The introgressed BC topologies have overall higher support values and are widely spread across the genome (referenced by the *A. thaliana* genome) (Supplementary Fig. 26b, c). These results are consistent with the study of Forsythe et al[58]. Again, Syn-MRL supported (B, (A,C)) (Supplementary Fig. 27).

We then analyzed the corresponding synteny profile patterns of the identified conserved low-copy gene orthogroups and found that in total, 95% of the orthogroups correspond to 'present in all' (5,343 orthogroups) and 'presence in all but the outgroup' (622

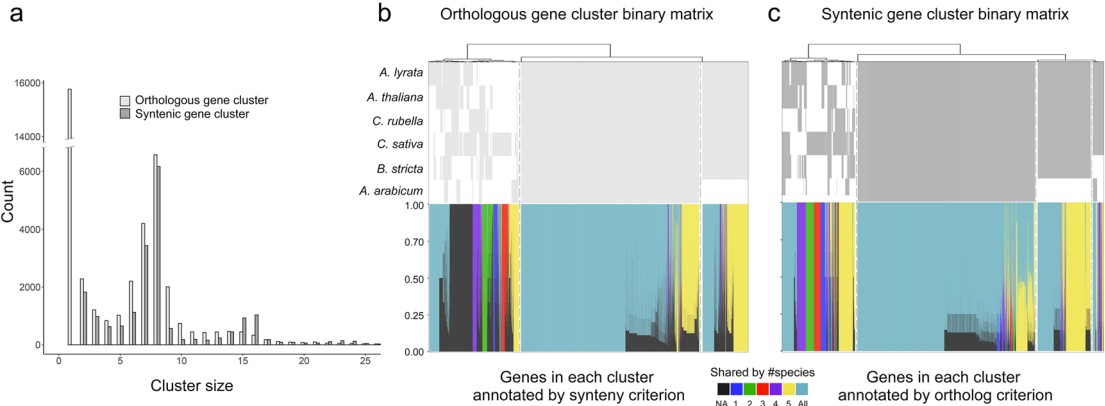

**Fig. 5 Comparison of gene clustering based on orthology and synteny of the ABC dataset (see text for details). a** Comparison of the sizes of orthogroups and synteny clusters. **b** (Upper) binary phylogenomic profiles for all orthogroups (size > 1) (clustered into three groups separated by dashed lines) and (bottom) corresponding synteny profiles (clustered by Jaccard distances within each group). We use the number of involved species to annotate profiles, note that one orthogroup can correspond to multiple synteny clusters, and vice versa. **c** (Upper) binary phylogenomic profilings of all synteny clusters (clustered into four groups separated by dash lines) and (bottom) corresponding orthogroup profiles (clustered by Jaccard distances within each group).

orthogroups) synteny profile patterns (Supplementary Data 3). This indicates that synteny profiles of the characterized conserved low-copy orthogroups (identified by gene content analysis) are not very informative for resolving phylogenetic relationships. To locate the 'informative' synteny profiles and understand their origin, we then compared the complete gene content matrix to all synteny clusters (Fig. 5). It is worth to mention that there is a considerable number of single-gene clusters (15,764 genes) that are not assigned into groups by gene orthology clustering (Fig. 5a). However, 9,623 of these genes (61.04%) are included in some synteny cluster, while the remaining 6,141 genes were not found in synteny clusters either (Supplementary Data 4). Such differences must be due to different input graphs (edges based on sequence similarity alone vs. edges based on sequence and neighborhood similarity) and clustering methods (MCL[60] vs Infomap[61], edge-weighted vs unweighted) being used in these two approaches. However, a comprehensive comparison regarding clustering accuracy and completeness would be necessary for future studies. Nevertheless, we found specific gene content profiles to be related to specific synteny profiles, and vice versa (leftmost clustered groups of the top panels of Fig. 5b, c). This means that specific orthogroups that have undergone differential gene gain/loss harbour useful signals for synteny-based phylogenetic reconstruction. However, commonly adopted phylogenetic reconstruction approaches based on conserved single-copy/low-copy gene markers would neglect such phylogenetic signals which can be especially useful in the scenarios where extensive introgression has occurred. In line with this, phylogenetic reconstruction based on the entire gene content matrix (size > 1) of the ABC dataset resolved the topology proposed by Forsythe et al.[58]. (Supplementary Fig. 28). However, for the 123 plant genomes, the reconstructed species tree based on gene content alone was apparently less accurate (Supplementary Fig. 29). As mentioned above the quality of orthology clustering can be a main concern especially for large data sets.

It is currently unknown whether ancestral introgressive hybridizations also occurred in related lineages and as such could provide a putative explanation for our findings concerning the positioning of *Arabis* in Brassicaceae and Pooideae in Poales in our synteny tree (Supplementary Fig. 21). We note that the phylogenetic position of *Arabis* has also been controversial[62,63]. A recent study identified shared genomic block associations between *Arabis* and *Aethionema* (which is sister to all other

Brassicaceae)[64]. Based on macrosynteny, the study of Walden et al. 2020 supports Arabideae as the clade diverging after the the divergence of *Aethionema* within Brassicaceae[64], which is consistent with our synteny tree (Fig. 4, Supplementary Fig. 21). Regarding Pooideae in Poales, the relationships regarding the PACMAD and the BEP clades[65] recovered by our synteny tree have—to our knowledge—rarely been reported (Fig. 4, Supplementary Fig. 21). However, the comparative phylogenomic large-scale gene family expansion and contraction analysis from the wheat genome project showed a similar clustering pattern of species based on gene family profiles, consistent with our result (Fig. 4a of International Wheat Genome Sequencing Consortium, 2018[66]). If the synteny tree were indeed true, it would reshape our understanding of the evolution of some of the most important crops including wheat, rice, and maize, as well as the origin of the $C_4$ lineages[67].

## Discussion
Phylogenetic trees serve many purposes and are an indispensable tool for the interpretation of evolutionary trends and changes. So far, abundant tools and sophisticated substitution models have been developed for molecular sequence alignment-based phylogenetic inference. In contrast, although it is well acknowledged that there is phylogenetic signal in gene order dynamics[4,68–72], efficient tools for analyzing phylogenomic synteny properties, reconstructing ancestral genomes, and inference of genome rearrangements, particularly for large data sets, remain scarce. In this study, we present an approach that bypasses the challenging combinatorial problem of inferring genome rearrangements and ancestral genome organization by integrating information on pairwise homologous genomic organization across a potentially large number of genomes in a network representation. As in classical orthogroup inference, we perform similarity searches across multiple genomes and cluster a network representation thereof, but our graph representation now also includes the genomic context (i.e., synteny information), and the resulting clusters are therefore defined by sequence level homology and structural homology at a microsynteny level. The idea developed in the present paper is that each cluster, whether broadly conserved, family- or lineage-specific, can readily be used as a phylogenetic signal. By reducing whole-genome syntenic comparisons to the synteny network representation, information with regard to the actual syntenic contexts of the genes is ignored

while only relations of shared syntenic contexts between genes are retained. By identifying clusters in the resulting synteny network, evolutionarily relevant homologous features of genome structure across species are efficiently captured while we are able to abstract from much of the complex features of genome evolution.

We have investigated the accuracy of our Syn-MRL approach using simulated data and empirical data sets (Figs. 2 and 3). We proposed a simple Markov model for the evolution of gene families with syntenic edges among family members ('gene family syntenic networks') in the context of a known phylogeny and observed good and consistent performance of Syn-MRL on simulated data sets under this model. At the moment, it is however largely unclear to what extent such a simple probabilistic model provides a reasonable model for empirical syntenic networks. We believe further research into probabilistic models for the evolution of these gene family syntenic networks is a promising avenue for new phylogenetic methods for the genomic era.

In this study, we did not perform inference under a dedicated model of syntenic network evolution like the one discussed above, but use the Mk model in a somewhat ad hoc approach reminiscent of MRL approaches. The Mk model is a Jukes-Cantor type model for discrete morphological data[73]. And we consider each column of the data matrix as an independently evolving character. The two-state binary encoding for each column represents the two groups of related species regarding that specific character. There is no further ordering or special weighing for the elements in the data matrix. The Mk model may arouse some concern regarding the symmetric two-state model (i.e., there is an equal probability of changing from state 0 to state 1 and from state 1 to state 0), as generally, time-reversible Markov models of evolution may not be ideal for the Syn-MRL approach. However, a first and important observation was that these models do seem to result in very reasonable well-supported phylogenies, in a similar vein as the MRL approach[30]. We hypothesize this is because, firstly, we may reasonably suspect that for our data, state 0 or 1 has the same probability of being the ancestral or derived state, as we can hypothesize that the emergence of a new lineage-specific synteny cluster and loss of another, are due to the same processes, e.g. transposition. Secondly, although the Mk model allows numerous state changes, in practice it yields trees with near-identical likelihood scores compared to a modified Mk model where character states can change only once or not at all[73]. This suggests that more biologically plausible non-reversible models, where for instance the re-emergence of a synteny cluster (a secondary $0 \rightarrow 1$ transition after the initial emergence and subsequent loss of the cluster) occurs at a different rate than the initial $0 \rightarrow 1$ transition, might not result in a substantially better fit. Moreover, it should be noted that similar binary models have already been used in ML analyses of ancestral reconstructions, such as the studies based on gene content[74,75], gene adjacencies[17,18], intron–exon structures[76,77], and morphological characters[78–80]. Nevertheless, we reiterate that the development of probabilistic models (like the model we use in our simulations) and inference algorithms for the evolution of syntenic networks may be a promising path to enable phylogenetic inference from structural features of whole-genome data.

Despite much progress in the last two decades, it is currently well-acknowledged that some phylogenetic relationships in plants remain especially controversial. For example, the relationships among monocots, eudicots, magnoliids, Ceratophyllaceae, and Chloranthales remain unsolved[25,48,53,81], while also the relationships among core rosids, asterids, Saxifragales, Vitales, Santalales, and Caryophyllales have been enigmatic[25,48–52]. We have compared our synteny tree with SA trees and state-of-art classifications (represented by APG (IV) and 1KP) in a general way

(Supplementary Figs. 21–24). Overall, our synteny tree showed great accuracy and congruence in the classification of the major lineages and clades. On the other hand, the synteny tree also provides alternative sister-group relationships for some of the recalcitrant clades mentioned higher. It should be noted that it is not our intention to argue that our tree is the 'true' tree, finally resolving the contentious relationships between some of the plant clades, but rather to provide a way to reconstruct and consider large-scale gene order-based phylogenetic trees, which may contribute to integrative phylogenetic analyses for challenging relationships.

The phylogenetic position of magnoliids has long been discussed[53], and even recently, based on whole-genome information of new magnoliid genomes, different sister-group relationships for magnoliids have been proposed[54–56,82]. Also, a sister-group relationship for magnoliids and monocots, as suggested in the current study, has been suggested before, based on different approaches and data[83–87]. Based on microsynteny, we have explored the relationships of magnoliids, monocots, and eudicots in more detail. Our synteny-based approach provides a means to investigate phylogenetic signal in the data matrix and trace those back to the genomic regions where differential gene order arrangements are located. To this end, we focused on the 'submatrix' of synteny profiles for magnoliids and extracted 15,424 magnoliid-associated synteny clusters (Fig. 6a). A hierarchical clustering of the phylogenomic profile of these synteny clusters showed 1107 synteny clusters that may be related to the grouping of magnoliids and monocots (Fig. 6a). To validate their contribution to the final ML tree, we first removed these signals from the entire matrix and reconstructed the phylogenetic tree, after which the species tree obtained favors magnoliids as sister to eudicots (BS = 100%) (Supplementary Fig. 30). To further understand the genomic distribution of specific genes in these clusters joining magnoliids and monocots, we reorganized the cluster profiles according to the chromosome gene arrangement of a magnoliid representative (Cinnamomum kanehirae)[54] (Supplementary Data 5). In doing so, we observed a number of 'signature' blocks consisting of specific anchor pairs that are shared by monocots and magnoliids with exclusion of eudicots (Supplementary Data 5-sheet 1, contexts with highlighted yellow rows). As an example, we highlight a synteny context of 15 genes where 8 genes (highlighted red) are only found in synteny between magnoliid and monocot genomes (Fig. 6b), with flanking genes generally conserved across angiosperms (highlighted in blue) (Fig. 6b, Supplementary Data 5-sheet2). However, an alternative explanation could be that the synteny context is lost in eudicots, from which it might be wrongly concluded that monocots and magnoliids share some derived characters and therefore share a common ancestry. Also, some synteny contexts are shared with early diverging eudicots (Ranunculales) from the 1107 clusters (Supplementary Data 5). Nevertheless, with more representative genomes of Chloranthales, Ceratophyllales, and early diverging angiosperms to be added into the analysis, a better resolution of the genomic rearrangements for magnoliids and related lineages could be obtained.

Apart from the contentious phylogenetic relationship of magnoliids with related lineages, evolutionary relationships of early-diverging lineages in superrosids or superasterids of core eudicots, such as Vitales, Saxifragales, Santalales, and Caryophyllales as suggested by microsynteny are strongly supported compared to competing hypotheses (as reflected by the AU test statistic). For example, our synteny-based phylogenies strongly support Vitales as early-diverging, right after Proteales (Nelumbo). Trees based on concatenation, MRL, and ASTRAL-CSSC trees in general also support Vitales as early diverging but also forming a sister-group relationship with

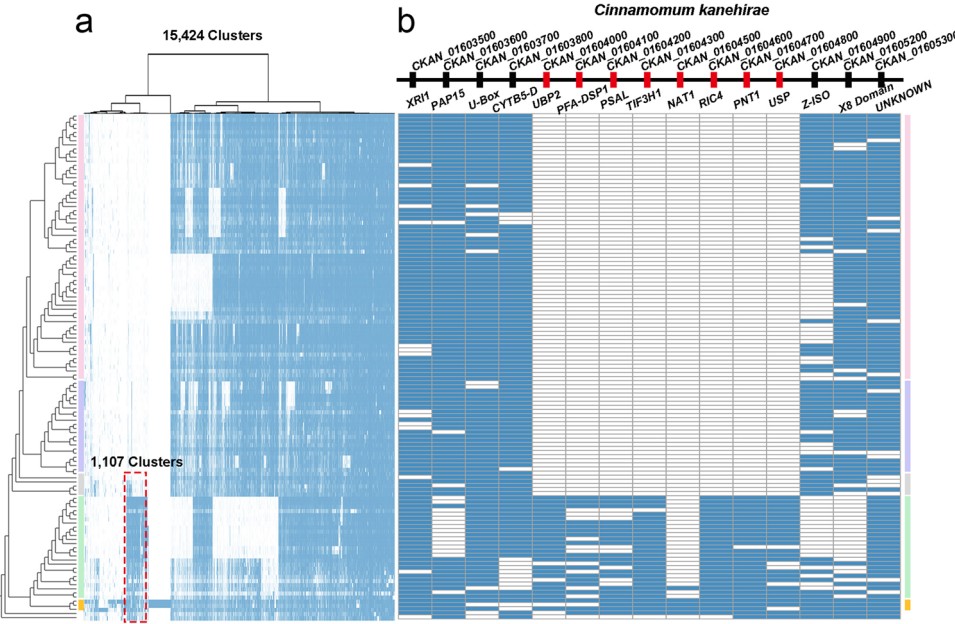

**Fig. 6 Magnoliids-associated signals and a representative example of phylogenetically informative microsynteny. a** Hierarchical clustering (method: ward.D) of 15,424 magnoliids-associate cluster profiles based on Jaccard distance. On the far-left, the synteny-based species tree is displayed (same as Fig. 4). Superrosids, superasterids, early diverging eudicots, monocots, and magnoliids are shaded in light-red, light-purple, light-grey, light-green, and light-yellow, respectively. 1107 clusters supporting a grouping of magnoliids and monocots (Supplementary Data 5). **b** One example from all supporting signals. A fifteen-gene context in the genome of *Cinnamomum kanehirae* (a magnoliid) shows eight neighboring genes (highlighted in orange) only present in magnoliids and monocot genomes, while the flanking genes (colored blue) are generally conserved angiosperm-wide.

Santalales (Supplementary Figs. 15–18, and 20). Several studies have already reported the positioning of Vitales as sister to the core eudicots or as early-diverging eudicots, based on mitochondrial genes or genomic data[88,89], concatenation of nuclear gene alignments[90], and large-scale transcriptome data[91,92]. The 1KP study also observed substantial gene-tree discordance for Vitales from analyses employing coalescent and supermatrix approaches, as well as plastid genomes[25]. It has been suggested that the gamma palaeohexaploidy at the root of the core-eudicots has involved hybridization (i.e., allopolyploidy)[93–96]. Thus phylogenetic incongruence of the relationships among abovementioned early diverging lineages in superrosids or superasterids of core-eudicots could potentially be attributed to ancestral hybridization(s).

The notion that species boundaries can be obscured by introgressive hybridization is increasingly accepted[59,97–100]. Thus, it seems plausible that some of the incongruences discussed above reflect introgression caused by (recent or more ancestral) hybridization. Using the ABC example, we showed that when introgression had occurred, phylogenetic reconstruction based on single-copy/low-copy gene markers can lead to well-supported branching patterns that do not reflect true species divergences. Since the focal point of synteny information is to reflect structural variation instead of gene sequence changes, it is plausible that genome-level synteny-based phylogeny inference may avoid the bias caused by (extensive) introgression in sequence-based species tree reconstruction and is more likely to reflect the true (or perhaps better, 'main') species branching order—provided that introgression remains possible after certain changes in genome structure. Furthermore, TE mobilization often follows hybrid speciation or introgression[101–103] which leads to gene transpositions, which may appear as phylogenetic signals in our data (explained in Fig. 1), and as such further strengthen patterns supporting the monophyly of the relevant clades. Nevertheless, in the case of true hybrid speciation, where a bifurcating species tree

does not exist, our method can be similarly unsuitable as other phylogenetic inference methods that do not allow phylogenetic network inference. In any case, the whole-genome synteny approach could provide unique opportunities towards developing phylogenetic inference methods that are more robust to introgressive hybridization, and further simulation-based studies should be able to shed more light on this possibility[104].

Here, we have presented a methodological roadmap to reconstruct species trees based on synteny information from large volumes of available whole-genome data, which can be applied to any set of genomes. However, it should be noted that our approach depends on the quality of genome assemblies and their gene annotations, which is the basis for synteny detection. Also, choosing appropriate parameters in synteny detection algorithms clearly presents a major challenge to any synteny-based phylogenetic approach. Thus, parameter settings for synteny detection should be tested and compared beforehand (see higher, 'Tests on empirical genome data sets'; here, we adopt the parameterization found to be most appropriate for the analysis of angiosperm genomes in a previous study[23]). One should also consider the evolutionary distance and genomic properties of the included genomes, as synteny conservation can be limited for distantly related species. For example, caution should be taken when comparing a gymnosperm or fern genome to an angiosperm genome because, in such comparisons, microsynteny might be inadequate due to extensive genome rearrangements. On the contrary, for highly similar genomes, our approach might simply not provide enough resolution due to the lack of informative rearrangements. However, in such cases, stricter synteny detection parameter settings, as well as consideration of gene orientation might help to increase the number of informative signals for resolving the tree topology.

To conclude, using a character matrix derived from a network representation of pairwise microsynteny relations, we here explored an approach inspired by MRL to reconstruct

phylogenies using genome structure data across a large set of angiosperm genomes. Our resulting synteny-based species tree shows high resolution and overall strong consistency with phylogenies of angiosperms obtained using more classical methods to infer tree topologies, although some notable differences were identified. We hope that our approach might offer a complementary way to consider and evaluate ambiguous phylogenetic relationships. Furthermore, as more and more high-quality genomes from underrepresented plant phyla are becoming available at increasing rates, we expect our approach to become more sensitive and informative in future applications.

## Methods

**Simulation**. To test the accuracy of the Syn-MRL approach for the reconstruction of species trees, we implemented a simple continuous-time Markov model of gene family evolution to simulate syntenic networks for gene families for a given species tree (Fig. 2). The model is a simple linear birth-death process embedded in the branches of the species tree, where genes evolve from the root to the tips with gene duplication, loss, and rearrangement events occurring at rate $\lambda$, $\mu$, and $\nu$ per gene respectively. Upon a speciation event, all extant genes are copied to the two daughter branches, and syntenic edges are created between pairs of genes that share a common ancestor in the parent branch. Upon a duplication event, a gene is added to the family in the branch where the duplication occurs, and each syntenic edge of the parental gene is copied to the new gene with probability $p_d$. Upon a rearrangement event, a gene loses each syntenic edge with probability $p_r$. The result of a single simulation is a 'gene family syntenic network', i.e. a syntenic network for a single gene family evolved through gene duplication, loss, and rearrangement dynamics along the species tree. Synteny *clusters* are then obtained by identifying connected components in the resulting gene family syntenic networks, which are then analyzed using the Syn-MRL approach described in detail below. We used two input species trees in our simulations. The 15 monocot species tree (*Oryza sativa, Sorghum bicolor, Zea mays, Brachypodium dystachion, Ananas comosus, Elaeis guineensis, Musa acuminata, Asparagus officinalis, Phalaenopsis equestris, Apostasia shenzhenica, Spirodela polyrhiza,* and *Zostera marina*) was based on TimeTree[105] with median node ages. The gene content matrix of the 15 species (generated by OrthoFinder[106] (version 2.4.0)) was used to estimated gene duplication ($\lambda$) and loss rates ($\mu$) using the DeadBird Julia package[107]. We assumed $\lambda = \mu$ (i.e., we estimate a single *turnover* rate), using a non-informative prior and a Geometric($p$) prior on the number of genes at the root of a gene family, with a Beta(3,1) prior on $p$. The marginal posterior mean value of $\lambda$ was used in all simulations. The 62 plant species tree was derived from the dated tree reported in Morris et al.[33]. IQ-TREE[108] (version 1.7-beta7) was used for ML tree inference (under the model 'MK + R + FO'), as well as the Robinson-Foulds distance calculation between the inferred trees and the input species tree. The model and simulation methods are implemented in the Julia programming language, and are available together with the species trees used in the simulation at https://github.com/arzwa/TaoNet.

**Empirical genome datasets**. Genome annotations of the species used in the data sets of YGOB[34], *Drosophila*[35], vertebrates[16], and yeast[16] were downloaded from public resources and processed (Supplementary Data 1). The Syn-MRL pipeline (for details see below) was applied to each data set. Synteny network construction, clustering, and tree inference were conducted respectively for each dataset under different settings of '-s' ($A_{min}$, the minimum number of anchor pairs required to call a synteny block) of MCScanX[36] (2, 3, 5, 7, 9, 11, and 13). Sequence alignment-based trees were reconstructed using IQ-TREE[108] (version 1.7-beta7) (under the model of 'JTT + R', -alrt 1000 -bb 1000) based on the concatenation of CSSC genes (genes in Conserved Single-copy Synteny Clusters, see below) identified for each data set – except for the yeast dataset where we used conserved single-copy genes identified by OrthoFinder[106] (version 2.4.0) instead, as no CSSC genes were identified (due to poor synteny conservation between the phylogenetic groups).

**Plant genome resources**. Reference genomes were obtained from public repositories, including Phytozome, CoGe, GigaDB, and NCBI. For each genome, we downloaded FASTA format files containing protein sequences of all predicted gene models and the genome annotation files (GFF/BED) containing the positions of all the genes. We modified all peptide sequence files and genome annotation GFF/BED files with corresponding species abbreviation identifiers. After constructing our synteny network database and clustering (see further), poor quality genomes could be relatively easily identified, and were removed from the database for further analysis (see further). After quality control, the final list of genomes used in the current study and related information for each genome can be found in Supplementary Data 2-Sheet1, genomes that were filtered out due to low contiguity were listed in Supplementary Data 2-Sheet2.

We acknowledge that our taxon sampling is limited to currently available high-quality genomes and thus many important lineages could not be included. For example, some important plant lineages still lack well-assembled genomes (such as for Dilleniales and Chloranthales), while others only have few representatives (such as Santalales and Saxifragales), and still others are relatively 'over-represented'

(such as Brassicales, Fabales, and Poales). For 'over-represented' orders, we kept all qualified genomes because this provides an opportunity to test the resolution and robustness of our approach at different levels (e.g. order-level, family-level, and species-level).

We manually downloaded each genome and checked the completeness of the annotation files. First, all candidate genomes were used for the synteny network construction. Next, we assessed genome quality from the phylogenomic profile matrix of all synteny clusters (where rows represent species/genomes and columns are clusters). Genomes with poor completeness and contiguity—indicated by lighter rows (for an example, see Fig. 5 of Zhao and Schranz, 2019[23], rows indicated by black arrows)—are removed from the microsynteny network. After this step, 123 fully sequenced plant genomes were used for further analysis (Supplemental Data 2). The overall sampling covered 31 orders and 52 different families of angiosperms.

**Pipeline for Syn-MRL**. Our synteny-based phylogenetic reconstruction approach includes four main steps, (1) phylogenomic synteny network construction, (2) network clustering, (3) matrix representation, and (4) maximum likelihood (ML) based tree inference. The synteny network construction consists of two main steps: first an all-vs.-all reciprocal sequence similarity search for all annotated proteomes was conducted using DIAMOND[109] (version v0.9.18.119), followed by MCScanX[36], which was used for pairwise synteny block detection. Parameter settings of MCScanX for angiosperm genome dataset have been tested and compared before;[23] here we adopt 'b5s5m25' (b: number of top homologous pairs, s: number of minimum matched syntenic anchors, m: number of max gene gaps), which has proven to be appropriate by various studies for the evolutionary distances among angiosperm genomes. To avoid large numbers of local collinear gene pairs due to tandem arrays, if consecutive homologs (up to five genes apart) share a common gene, homologs are collapsed to one representative pair (with the smallest E-value). Further details regarding phylogenomic synteny network construction can be found in Zhao and Schranz (2018) and a tutorial available in the associated GitHub repository (https://github.com/zhaotao1987/SynNet-Pipeline/wiki). In the resulting synteny network, nodes are genes in syntenic blocks, while edges connect syntenic anchor pairs. For our work, the entire synteny network summarizes information from 7,435,502 pairwise syntenic blocks, and contains 3,098,333 nodes and 94,980,088 edges.

The entire synteny network is clustered for further analysis. We used the Infomap algorithm (version 0.20.0) for detecting synteny clusters within the map equation framework[61] (https://github.com/mapequation/infomap). We have discussed before why Infomap is more appropriate for clustering phylogenomic synteny networks[23]. We used the two-level partitioning mode with ten trials (–clu -N 10 –map -2). The network was treated as undirected and unweighted. Resulting synteny clusters vary in size and composition, which is associated with synteny either being well-conserved or rather lineage-/species-specific. A typical synteny cluster comprises of syntenic genes shared by groups of species, which precisely represent phylogenetic relatedness of genomic architecture among species (Fig. 1). Here, we clustered the entire synteny network into 137,833 clusters. A cluster phylogenomic profile records the number of nodes in a given cluster for each species, and the collection of phylogenomic profiles is summarized in a data matrix where rows and columns represent species and clusters respectively (Fig. 1). For phylogenetic inference, the matrix was then reduced to a binary presence-absence matrix to obtain the final synteny matrix (Fig. 1).

Tree inference was performed using maximum likelihood with implemented in IQ-TREE[108] (version 1.7-beta7), using the Mk+R + FO model. (where "M" stands for "Markov" and "k" refers to the number of states observed, in our case, k = 2). The +R (FreeRate) model was used to account for site-heterogeneity, and typically fits data better than the Gamma model for large datasets[110,111]. State frequencies were optimized by maximum-likelihood (by using '+FO'). We generated 1000 bootstrap replicates for the Shimodaira-Hasegawa[112] like approximate likelihood ratio test (SH-aLRT), and 1000 ultrafast bootstrap (UFBoot) replicates (-alrt 1000 -bb 1000)[113].

For gene content-based phylogenetic reconstruction (Supplementary Fig. 29), we obtained orthogroups for the 123 plant genomes using OrthoFinder[106] (version 2.4.0). The orthogroup phylogenomic profile matrix consisted of 24,727 orthogroups and was converted to a binary presence/absence matrix. IQ-TREE[108] (version 1.7-beta7) was used for ML tree inference, under the same model and settings as used for Syn-MRL.

**Sequence alignment-based phylogenetic reconstruction**. For sequence-based phylogenetic inference, we employed three commonly used approaches, namely a supermatrix (also called superalignment or concatenation) approach, a reconciliation approach based on the multispecies coalescent (MSC), and a supertree approach using matrix representation with likelihood (MRL). For each of these, we used two sets of whole-genome derived gene markers independently, namely BUSCO genes (Benchmarking Universal Single-Copy Orthologs)[114] and CSSC genes (Conserved Single-copy Synteny Clusters).

We used the following criterion for characterizing CSSC genes: median number of nodes across species < 2, present in ≥ 111 (90%) genomes, and presence within Poaceae, monocots (except Poaceae), Asterids, Rosales, Brassicaceae, and Fabaceae must ≥ 50%. BUSCO analysis (v3.0, embryophyta_odb9, with 1440 profiles)

identified a total of 1438 conserved single-copy genes from the 123 angiosperm genome sequences, compared to 883 identified as CSSC. Multiple sequence alignments were performed using MAFFT[115] (version 7.187). First, the alignments were trimmed using trimAl[116] (version v1.4.rev15) through heuristic selection of the automatic method (-automated1). Second, sequences with less than 50% residues that pass the minimum residue overlap threshold (overlap score: 0.5) were removed (-resoverlap 0.5 -seqoverlap 50). Alignment concatenation was conducted by catfasta2phyml (https://github.com/nylander/catfasta2phyml) The length of our concatenated gene sequence alignments of BUSCO and CSSC genes were 591,196 and 341,431 amino acids, respectively.

Maximum-likelihood analyses were conducted using IQ-TREE[108] (version 1.7-beta7). For sequence-based tree construction, we used the JTT + R model for protein alignments, both for the construction of trees based on single alignments, as well as for the concatenated sequence alignments. For all trees, we performed bootstrap analysis with 1000 bootstrap replicates (SH-aLRT and UFBoot (-alrt 1000 -bb 1000)).

ASTRAL-Pro[117] was used for the tree summary approach based on the multi-species coalescent to infer the species tree from the 1438 BUSCO gene trees and 883 CSSC gene trees. ASTRAL-Pro is the latest update of ASTRAL, which can now account for multi-copy trees.

For the MRL supertree analysis, 1438 BUSCO gene trees and 883 CSSC gene trees were used as two independent data sets. We encoded all splits (bipartitions) with ≥ 85% UFBoot support among all the trees into the data matrix. Thus, each column in the matrix represents a well-supported bipartition (this coding is similar to the "Baum-Ragan" coding method (0, 1,?)[118], but without question marks because '?' was originally designed for missing taxa as trees were multi-sourced). The dimensions of the matrices are $123 \times 139{,}538$ for the BUSCO gene trees, and $123 \times 102{,}617$ for the CSSC gene trees. We used the same binary model (Mk + R + FO) in IQ-TREE, and parameter settings were identical to those described above for the Syn-MRL supercluster analysis.

To assess whether alternative sister-group relationships of certain plant clades could be statistically rejected given the synteny matrix, we performed approximate unbiased (AU) tests[57], as implemented in IQ-TREE[108] (version 1.7-beta7), under the 'MK + R + FO' model, with 10,000 replicates.

**Analysis of the ABC dataset.** The ABC dataset refers to the six Brassicaceae genomes representing Clade A (*A. lyrata* and *A. thaliana*), Clade B (*B. stricta*), Clade C (*C. rubella* and *C. sativa*), and the outgroup (*A. arabicum*). Orthogroup inference was conducted using OrthoFinder[106] (version 2.4.0) (with default inflation parameter: -I = 1.5). For sequence alignment based tree reconstructions, we identified 11,611 conserved low-copy orthogroups instead of single-copy orthogroups (criterion: present in all six species and the average copy number < 2) since *C. sativa* has undergone whole-genome triplication relative to *C. rubella*[119]. Phylogenetic reconstruction was performed for genes in each orthogroup, using the same tools and methods as used for plants (see upper). We filtered the resulting trees by topology, where we retained those trees consistent with the monophyly of each of the three clades and that could be properly rooted by a monophyletic *A. arabicum* group, resulting in a set of 6,306 trees. For these trees, we counted the support for each of the three competing topologies (Supplementary Data 3). The script for analyzing tree topologies was adapted from the study of Forsythe et al.[58]. In parallel, Syn-MRL was applied to the ABC dataset. Phylogenomic profiles were obtained for synteny clusters and orthogroups (Fig. 5b, c). Each of the synteny clusters and orthogroups was annotated by its number of represented species. We compared the two groupings by analyzing the compositions of each synteny cluster or orthogroup according to each other's criterion. For example, genes in one orthogroup can be clustered as one synteny cluster, or can be split into several synteny clusters, and vice versa. Genes present in one dataset can be absent in another (NAs on Fig. 5). Gene content-based phylogenetic reconstruction was performed using the same method as described for the 123 plant genomes above.

**Reporting summary**. Further information on research design is available in the Nature Research Reporting Summary linked to this article.

## Data availability
Data sets used in this study are available at DataVerse[120] (https://doi.org/10.7910/DVN/7ZZWIH). This includes all annotated protein sequences in FASTA format of each genome, entire synteny network database (edge list), network clustering result, trimmed alignments and corresponding phylogenetic trees of BUSCO and CSSC genes, bipartitions with support values for each tree, and binary data matrices.

## Code availability
Scripts for synteny network construction, network clustering, and phylogenomic profiling are available at Github (https://github.com/zhaotao1987/SynNet-Pipeline). Related code (for preparing the data matrices for phylogenetic reconstructions) and software parameters are available at Github (https://github.com/zhaotao1987/Syn-MRL). The gene family syntenic network simulation program is available at Github (https://github.com/arzwa/TaoNet).

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

## Acknowledgements
T.Z. is supported by the start-up funding provided by Northwest A&F University (No. Z111022001). A.Z. is supported by a PhD fellowship of the Research Foundation Flanders (FWO). Z.L. is supported by a postdoctoral fellowship from the Special Research Fund of Ghent University (BOFPDO2018001701). Y.V.d.P. acknowledges funding from the European Research Council (ERC) under the European Union's Horizon 2020 research and innovation programme (Grant agreement no. 833522) and from Ghent University (Methusalem funding, BOF.MET.2021.0005.01).

## Author contributions
Y.V.d.P., M.E.S, and J.X. conceived the idea. T.Z., A.Z., Y.V.d.P. designed the study. T.Z. and S.K. performed the analysis. A.Z. implemented the simulation program. T.Z., J.X., Y.V.d.P, Z.L., M.E.S., and A.Z. analyzed data. T.Z., A.Z., and Y.V.d.P. wrote the manuscript. All authors discussed the results and commented on the manuscript.

## Competing interests
The authors declare no competing interests.
