## [Peer Review File · Nature Communications]

Reviewers' Comments:

Reviewer #1:

Remarks to the Author:

Zhao et al present an interesting analysis of microsynteny data applied to phylogenetic analysis of relationships among major flowering plant lineages. The analysis is largely consistent with published results from analyses of gene sequences indicating that there is reliable phylogenetic signal in the synteny cluster matrix that the authors have constructed. However, emphasis is placed on instances where synteny-based and gene sequence-based analyses are discordant. These differences are interesting, but difficult to interpret without more information on the synteny-based method and how it may be impacted by variation in the processes that maintain or disrupt synteny.

Major concerns:

1. While Figure 1 provides a very nice high-level view of the steps employed to produce a presence/absence data matrix, the simple Markov model used to analyze the matrix is not represented on the figure. The description of the Mk model and its possible limitations isn't presented until the discussion (starting on line 250). Possible issues associated with time-reversibility (allowing gain and loss of synteny clusters) are discussed, but the only assessment of the method's accuracy is general concordance with published sequence-based estimates of angiosperm relationships. No other assessment (e.g. using other benchmark data sets or simulated data) is performed.
2. Without such benchmarking of a new method, interpretation of results – including support values – is difficult. If the Mk model mis-specifies the process of synteny gain and/or loss, the inference method could be statistically inconsistent. Given the number of synteny clusters analyzed (137,833) the possibility of statistical inconsistency should be considered.
3. I may have missed it, but I did not find a clear description of how ancestrally polyploid genomes are handled. Is a cluster is scored as present if any of the subgenomes has the cluster? I wonder if the Mk model would estimate different gain/loss transition probabilities on branches with and without WGDs
4. The assertion that synteny based phylogenetic analyses may be less influenced by interspecific gene flow than sequenced-based phylogenomic analyses (starting at line 355) is interesting, but could be better supported with simulations. Recent analysis of sunflower genomes indicate that large genomic segments can be introgressed (Todesco et al. 2020). What proportion of genes were introgressed and causing problems for the Brassicales identified by Forsythe et al.?

Minor point: I don't follow the relevance of the columbine genome analysis by Aköz & Nordborg in the context of the placement of *Vitis* (starting a line 347). The conclusions of Aköz & Nordborg are controversial (e.g. see Xie et al. 2020, <https://doi.org/10.1038/s41438-020-0328-y>). In any case, how does the timing of a WGD in the evolution of the ancestral core-eudicot hexaploidy influence the placement of Vitales? Could a low rate of synteny loss within the Vitales lineage be responsible for its placement in the Syn-MRL analysis?

In my opinion the authors describe an interesting analysis and Syn-MRL could be of increasing utility as the number of chromosomal genome sequence is rapidly increasing. Like the other gene-order methods cited by the authors, however, their method should be described in more detail and assessed using multiple benchmarks and/or simulated datasets. If more comprehensive validation of the Syn-MRL method is outside the scope of the submitted manuscript, the findings with respect to the angiosperm phylogeny could simply be reported without assertions about accuracy.

Reviewer #2:

Remarks to the Author:

This submission is an interesting attempt to construct a phylogeny from genome arrangement data at a relatively large scale. The authors explore and interpret the few differences between a 123 angiosperm phylogeny made from gene sequences and gene arrangements.

For gene sequences, the phylogeny is constructed with standard methods. As the authors point, there is no such standard method for synteny, so they use a recently developed one, made from the construction of a binary matrix with synteny data, and its processing by a standard ML method.

On one side the method is a bit ad-hoc, made of several arbitrary choices. The Markov model of gains and losses of syntenic genes does not correspond to specific evolutionary events, which could be gain/loss of a gene or gain/loss of synteny. This won't help understanding evolutionary scenarios and gaining knowledge from such phylogenies.

On the other side, the results are honestly presented, and the discussion on the methodology and on the results is detailed, interesting, and raises deep questions about genome evolution.

I am in favor of publishing this submitted manuscript because it is, in my impression, a good step in the very difficult problem of inferring phylogenies with rearrangements.

That said I have a few comments that could improve the manuscript.

1/ It is true that phylogenetics with syntenic data is in its infancy. However the state of the art seems to me a bit more advanced than what acknowledged in this submission. For example the ML method with a pre-constructed binary matrix is already in the cited reference 11, and was described in several articles, like

Moret et al, Rearrangements in phylogenetic inference: Compare, model, or encode? *MAGE* 2013

Lin and al Maximum likelihood phylogenetic reconstruction from high-resolution whole-genome data and a tree of 68 eukaryotes *PSB* 2013

along with discussions on the best compromise between the complexity of the data and the complexity of the algorithms.

2/ The method seems to mix the signal of presence/absence of a gene, and its genomic context. As such it is related to the gain/loss models in phylogeny which, contrarily to genome rearrangements, are well developed, well known, methodologically involved and highly discussed, and gene presence/absence is frequently used on eukaryotic genomes when the signal from sequences misses. For a single, out of many, example, see

Pisani et al, Genomic data do not support comb jellies as the sister group to all other animals *PNAS* 2015

These should be included in the discussion, if acknowledged by the authors that it is relevant (I do not ask that these papers are cited, none of them is mine, I just feel that what they report miss in the present discussion).

In particular it could enrich the discussion on the gain/loss parameters, because these aspects are worked out in this context.

3/ I found the discussion on the position of the magnoliids interesting but not very convincing. If I understand well, the question is how to resolve the quadruplet (Magnoliids, Monocots, Eudicots, Outgroup), and the presented example shows a character present in three of them, and absent from one, so without apparent information to resolve the quadruplet. So I did not understand why it is presented as a support for a (Ma,Mo) clade, although I understood that the signal was probably there, because removing it changes the phylogeny. However I felt that why it is there is not captured by the article.

4/ The discussion about the brassicaceae phylogeny is also very interesting but would deserve some clarification. First it is written that according to reference 56, the "true" tree would be different from the tree supported by a majority of genes. It would be interesting to know how the true tree is known if the genes support another one (I suppose it is in the reference 56 but the argument misses in the present discussion), and to know what the true tree represents if genes follow another scenario. The

authors are precautionous, they put quotes when using "true" and discuss a bit this notion, but it would be interesting to discuss this not from the point of view of the truth, but from the point of view of the processes. If a syntenic tree is closer to the species tree, does it mean that synteny is harder to introgress than single genes? Can we deduce something on the hybridization or introgression processes? Can synteny be viewed as an good alternative to sequences for phylogenetics when the sequence signal is blurred by transfers or introgressions?

And finally, minor remarks and probable typos:

- there is a "may related" which I would simply transform to "related"
- in Supp Fig 8, it would be useful to write which tree corresponds to which method
- In the article it is written that a tutorial for the method is available at <https://github.com/zhaotao1987/SynNet-Pipeline> but I could not find it, the link leads to to the code.
- "Syntenic genes" is not explained enough on line 502. Does it mean "a link between two homologous genes when they appear in a pairwise synteny block"? This is how I interpreted it but I did not see it explicited and it is crucial to understand the method.

Reviewer #3:

Remarks to the Author:

This study uses synteny information to produce a phylogenetic tree that is largely congruent with the APG classification, which is almost entirely based on plastid information. Clearly the novel method retrieves results that are reliable, if judged from the perspective of congruence with other results. There are differences, but it is difficult to determine if this is a methodological problem or if the other data and methods used gave unreliable results. My opinion is that this paper tries to do too much in a format that does not allow a thorough examination of these novel methods. I suggest submission to journals where these methodological matters can be more thoroughly evaluated before launching into results, especially those that differ from previous studies. I would also prefer to have results compared to those from datasets composed of nuclear gene data rather than just the APG (largely plastid) results. The differences could be just plastid versus nuclear trees. Fundamentally, I wonder if it is strictly necessary to involve such complexity in methods to obtain a phylogenetic tree when other datasets/tree building methods have already been found to perform well. Given that the results are largely congruent with the established ideas of general angiosperm relationships and the differences focus mostly on results that have had a history of being problematic, this paper does not add a great deal to the literature about phylogenetic results. Whether the methods are reliable or problematic in some respects needs a lot more attention before proclaiming it to be useful.

Response to the reviewers:

REVIEWER COMMENTS

Reviewer #1 (Remarks to the Author):

Zhao et al present an interesting analysis of microsynteny data applied to phylogenetic analysis of relationships among major flowering plant lineages. The analysis is largely consistent with published results from analyses of gene sequences indicating that there is reliable phylogenetic signal in the synteny cluster matrix that the authors have constructed. However, emphasis is placed on instances where synteny-based and gene sequence-based analyses are discordant. These differences are interesting, but difficult to interpret without more information on the synteny-based method and how it may be impacted by variation in the processes that maintain or disrupt synteny.

Major concerns:

1. While Figure 1 provides a very nice high-level view of the steps employed to produce a presence/absence data matrix, the simple Markov model used to analyze the matrix is not represented on the figure. The description of the Mk model and its possible limitations isn't presented until the discussion (starting on line 250). Possible issues associated with time-reversibility (allowing gain and loss of synteny clusters) are discussed, but the only assessment of the method's accuracy is general concordance with published sequence-based estimates of angiosperm relationships. No other assessment (e.g. using other benchmark data sets or simulated data) is performed.

Response: We understand and appreciate the reviewer's comments and concerns about the validity of the approach and model we used. As you will see in the revised manuscript, we have done substantial work on simulations and empirical datasets (new Figures 2 and 3 in the revised version of our manuscript) to test the accuracy and overall applicability of our method.

Briefly, for simulations we specified a continuous-time Markov model for the evolution of gene families along a species tree with syntenic edges between genes and simulated 'gene family syntenic networks' in stages across a given dated species tree. For benchmark datasets, we applied our Syn-MRL approach to four empirical data sets, including 19 genomes used in YGOB (Yeast Gene Order Browser), 12 *Drosophila* genomes, 13 vertebrate genomes, and another set of 21 yeast genomes. Overall, both simulations and the use of empirical data sets show our method to be accurate and consistent for many but not all data sets. Of course, it remains questionable whether the model used for simulating gene family syntenic networks is reasonably realistic, and we mention this in the text. However, we believe the model we employed for our simulations, although crude, might be a first step towards more relevant probabilistic models for phylogenetic inference from genome structure

information. In addition, we have greatly revised and restructured the text, and now the essence of our approach appears earlier in Results, as the reviewer suggested.

2. Without such benchmarking of a new method, interpretation of results – including support values – is difficult. If the Mk model mis-specifies the process of synteny gain and/or loss, the inference method could be statistically inconsistent. Given the number of synteny clusters analyzed (137,833) the possibility of statistical inconsistency should be considered.

Response: As we have replied under Question 1, we have performed simulations and applied our Syn-MRL approach to empirical data sets. At least under the birth-death model of gene family syntenic network evolution we employ - which is akin to widely used models for gene family evolution), the method is statistically consistent for the parameters examined – the accuracy increases with the number of gene families (up to 50k) and clusters (new Fig. 2b). This is all well explained in the revised version of our manuscript.

3. I may have missed it, but I did not find a clear description of how ancestrally polyploid genomes are handled. Is a cluster is scored as present if any of the subgenomes has the cluster? I wonder if the Mk model would estimate different gain/loss transition probabilities on branches with and without WGDs.

Response: We have made minor changes to Figure 1 and its legend to better illustrate and explain such scenarios. In Figure 1, Species 2 and 3 are exemplified genomes that have undergone WGDs. The presence of a gene in the data matrix is only dependent on whether this gene can be syntenic to another gene (from its own genome or with another genome). So, it does not need to be present in all subgenomes for a polyploid genome. For example, in Cluster 1, although one gene was retained for the polyploid Species 2, it is also considered as present (in the data matrix) for the gene family (colored in orange) in Cluster 1.

For the second question, the Mk model by default assumes equal transition probabilities of the different states. However, according to our empirical observations, species or clades with extra WGDs are often resolved with much longer branch lengths (which means more synteny changes), see for instance the genomes of *Brassica*, *Gossypium*, and Poales. In other words, for flowering plants, WGD seems to have led to more lineage-specific synteny clusters, aiding in phylogenetic reconstruction.

4. The assertion that synteny based phylogenetic analyses may be less influenced by interspecific gene flow than sequenced-based phylogenomic analyses (starting at line 355) is interesting, but could be better supported with simulations. Recent analysis of sunflower genomes indicate that large genomic segments can be introgressed (Todesco et al. 2020). What proportion of genes were introgressed and causing problems for the Brassicales identified by Forsythe et al.?

Response: We have added new analyses to the manuscript specifically addressing this question (represented by new Figure 5). For details, please refer to the main text. Briefly, we used six genomes representing Clade A (*Arabidopsis thaliana* and *A. lyrata*), Clade B (*Bochera stricta*), Clade C (*Capsella rubella* and *Camelina sativa*), and the outgroup (*Arabicum arabicum*) in our analysis to represent the introgression reported by Forsythe et al. We refer to this as the ABC data set in our study.

We performed Syn-MRL for this dataset, and in parallel we built phylogenetic trees using conserved low-copy gene families (orthogroups). Again, Syn-MRL resolved the species tree as (B,(A,C)) (AC topology). We sorted 11,611 gene trees and summarized the support for different topologies. Consistent with the results of Forsythe et al., we found that a majority of filtered gene trees supported the (A,(B,C)) topology (58.9%), which is supposed to be the reported introgressed topology (Forsythe et al 2020). This observed proportion for the BC topology is lower than what was reported in Forsythe et al. (87.6%) but note that the genome data sets being analyzed are not the same. Thus, at this step, we recovered the results reported by Forsythe et al.

We then analyzed the corresponding synteny profile patterns of the identified conserved low-copy gene orthogroups and interestingly, in total, 95% of the orthogroups correspond to 'presence in all' (5,343 orthogroups) or 'presence in all but the outgroup' (622 orthogroups) synteny profile patterns on a synteny matrix, indicating that synteny profiles for these genes are not very informative in solving phylogenetic relationships.

We then located 'informative' signals from the synteny matrix by comparing the synteny matrix and the gene content matrix. These 'informative' synteny profiles correlate with gene orthogroups that have undergone differential gain and loss. However, commonly-adopted sequence alignment approaches (based on conserved single-copy/low-copy gene markers) often have overlooked such phylogenetic signals, which can be especially useful in the context of introgression. As we show, our whole-genome synteny-based method can be a good complement in such scenarios.

Related results for this part of analysis include Figure 5, Supplementary Figures 26-29, and Supplementary Tables 4-5.

Minor point: I don't follow the relevance of the columbine genome analysis by Aköz & Nordborg in the context of the placement of *Vitis* (starting a line 347). The conclusions of Aköz & Nordborg are controversial (e.g. see Xie et al. 2020, <https://doi.org/10.1038/s41438-020-0328-y>). In any case, how does the timing of a WGD in the evolution of the ancestral core-eudicot hexaploidy influence the placement of *Vitales*? Could a low rate of synteny loss within the *Vitales* lineage be responsible for its placement in the Syn-MRL analysis?

Response: We have rewritten the corresponding part. According to the context, what we meant is that the hybrid(ization) origin of core-eudicots potentially explains current well-acknowledged controversial

relationships regarding Vitales, Santalales, Saxifragales, and Caryophyllales. They all belong to early-diverging groups of either superrosids or superasterids, which can be 'rogue' taxa if hybridization exists.

The dispute of Aköz & Nordborg's work is focusing on the timing of the WGD event in the early-diverging eudicot *Aquilegia*, whether the WGD is shared by all eudicots (Aköz & Nordborg) or it is *Aquilegia*-specific (Shi & Chen, 2020; Xie et al. 2020). However, it is relatively well accepted that the paleohexaploidy (gamma event) shared by core-eudicots has involved hybridization (Lyons et al. 2008; Jiao et al. 2012; Ming et al. 2013). We have rewritten the text to clarify our point.

Regarding the reviewer's second question: we think it is a very interesting concern. It is indeed possible that the positioning of Vitales is due to a low rate of synteny loss. We would like to better test this with simulations and explicit models in future work. However, based on the analyses we have already conducted, it is also worth noting that besides the synteny method, five out of six of the SA (sequence alignments) based trees also supported Vitales as early-diverging eudicot (Supplementary Figs. 15-20).

In my opinion the authors describe an interesting analysis and Syn-MRL could be of increasing utility as the number of chromosomal genome sequence is rapidly increasing. Like the other gene-order methods cited by the authors, however, their method should be described in more detail and assessed using multiple benchmarks and/or simulated datasets. If more comprehensive validation of the Syn-MRL method is outside the scope of the submitted manuscript, the findings with respect to the angiosperm phylogeny could simply be reported without assertions about accuracy.

Response: We acknowledge the reviewer's comments and have done substantial analyses as suggested to overcome the limitations. With the updated manuscript, we hope the reviewer's concerns have been addressed.

References:

- Shi, T., Chen, J. A reappraisal of the phylogenetic placement of the *Aquilegia* whole-genome duplication. *Genome Biol* 21, 295 (2020).
- Xie, J., Zhao, H., Li, K. et al. A chromosome-scale reference genome of *Aquilegia oxysepala* var. *kansuensis*. *Hortic Res* 7, 113 (2020).
- Lyons, E., Pedersen, B., Kane, J. & Freeling, M. The value of nonmodel genomes and an example using SynMap within CoGe to dissect the hexaploidy that predates the rosids. *Tropical Plant Biology* 1, 181-190 (2008).
- Jiao, Y. et al. A genome triplication associated with early diversification of the core eudicots. *Genome Biol* 13, 1-14 (2012).
- Aköz, G. & Nordborg, M. The *Aquilegia* genome reveals a hybrid origin of core eudicots. *Genome Biol* 20, 256 (2019).
- Ming, R. et al. Genome of the long-living sacred lotus (*Nelumbo nucifera* Gaertn.). *Genome Biol* 14, 1-11 (2013).

Reviewer #2 (Remarks to the Author):

This submission is an interesting attempt to construct a phylogeny from genome arrangement data at a relatively large scale. The authors explore and interpret the few differences between a 123 angiosperm phylogeny made from gene sequences and gene arrangements.

For gene sequences, the phylogeny is constructed with standard methods. As the authors point, there is no such standard method for synteny, so they use a recently developed one, made from the construction of a binary matrix with synteny data, and its processing by a standard ML method.

On one side the method is a bit ad-hoc, made of several arbitrary choices. The Markov model of gains and losses of syntenic genes does not correspond to specific evolutionary events, which could be gain/loss of a gene or gain/loss of synteny. This won't help understanding evolutionary scenarios and gaining knowledge from such phylogenies.

On the other side, the results are honestly presented, and the discussion on the methodology and on the results is detailed, interesting, and raises deep questions about genome evolution.

I am in favor of publishing this submitted manuscript because it is, in my impression, a good step in the very difficult problem of inferring phylogenies with rearrangements.

Response: We appreciate the positive comments from the reviewer. With respect to the concerns about our method, we have significantly expanded work on simulations and tested additional empirical data sets to further assess and validate the approach and the model we used. This is all added to the revised version of our manuscript (see also our responses to the questions of reviewer 1).

That said I have a few comments that could improve the manuscript.

1/ It is true that phylogenetics with syntenic data is in its infancy. However the state of the art seems to me a bit more advanced than what acknowledged in this submission. For example the ML method with a pre-constructed binary matrix is already in the cited reference 11, and was described in several articles, like

Moret et al, Rearrangements in phylogenetic inference: Compare, model, or encode? MAGE 2013
Lin and al Maximum likelihood phylogenetic reconstruction from high-resolution whole-genome data and a tree of 68 eukaryotes PSB 2013
along with discussions on the best compromise between the complexity of the data and the complexity of the algorithms.

Response: We have updated the Introduction in the revised manuscript to better summarize the developments in this field and have now cited related papers exploiting ML and binary matrix for phylogenetic reconstructions.

However, it is worth mentioning that although our Syn-MRL approach looks similar in form to an earlier published method (Lin et al 2013), essential differences exist in data preparation, meaning, and the composition of our binary matrix. The method by Lin et al (2013) is based on a concatenation of gene adjacencies and gene content, the data they used was from the eGOB database, which contains pre-calculated gene adjacencies and gene content data (Lopez et al. 2011). The 68 genomes dataset contains species from Amoebozoa, Euglenozoa, Viridiplantae, Metazoa, and Fungi, etc. The evolutionary distances and divergence times across and within these phylogenetic groups were huge. In the paper of Lin et al. (2013), the authors mentioned that the tree reconstructed from the adjacency matrix only (without gene content matrix) is very poor. Apparently, the gene content matrix contributed more to their final reconstructed tree. While it is plausible to use the gene-content matrix for phylogenetic reconstruction based on 'whole-genome data' (as in Lin et al., 2013), it is doubtful that this study should be regarded as a successful large-scale application of synteny information to phylogenetic reconstruction. Also, plant genomes are featured by prevalent whole-genome duplications and massive fractionations, thus we are skeptical about the actual resolution of using direct gene by gene adjacencies for plant genomes. Such a challenge has been formulated and discussed before (Sankoff et al. 2013).

References

Lopez MD, Samuelsson T: eGOB: eukaryotic Gene Order Browser. *Bioinformatics*.2011; 27:1150-1151.

Sankoff, David, and Chunfang Zheng. "Fractionation, rearrangement, consolidation, reconstruction." *Models and Algorithms for Genome Evolution*. Springer, London, 2013. 247-260.

2/ The method seems to mix the signal of presence/absence of a gene, and its genomic context. As such it is related to the gain/loss models in phylogeny which, contrarily to genome rearrangements, are well developed, well known, methodologically involved and highly discussed, and gene presence/absence is frequently used on eukaryotic genomes when the signal from sequences misses. For a single, out of many, example, see Pisani et al, Genomic data do not support comb jellies as the sister group to all other animals PNAS 2015. These should be included in the discussion, if acknowledged by the authors that it is relevant (I do not ask that these papers are cited, none of them is mine, I just feel that what they report miss in the present discussion). In particular it could enrich the discussion on the gain/loss parameters, because these aspects are worked out in this context.

Response: We appreciate the suggestion and have now included the use of such information to our Discussion. Phylogenetic reconstructions based on gene content (presence/absence matrix) do share great similarities with our Syn-MRL approach, especially in the form of the binary input matrix and the binary model being used. We were first inspired by the MRL/MRP supertree method, which is used to summarize branching patterns of trees, and our method can be regarded as a supercluster approach

that can be used for ML estimation of all synteny cluster profiles (a synteny cluster is a smallest unit of phylogenetic signal based on genomic structural homology across/within genomes). Gene content-based methods for phylogenetic reconstruction is indeed highly relevant to our work, and, additionally, can provide an extra foundation for our approach. Please note that in our revised paper, we include comparisons to inference from gene content data alone.

3/ I found the discussion on the position of the magnoliids interesting but not very convincing. If I understand well, the question is how to resolve the quadruplet (Magnoliids, Monocots, Eudicots, Outgroup), and the presented example shows a character present in three of them, and absent from one, so without apparent information to resolve the quadruplet. So I did not understand why it is presented as a support for a (Ma,Mo) clade, although I understood that the signal was probably there, because removing it changes the phylogeny. However, I felt that why it is there is not captured by the article.

Response: First, we have to admit that although a sister relationship was supported for magnoliids and monocots (BS = 95%) in our analysis (Fig. 4), alternative topologies cannot be ruled out from the data matrix perspective (Supplementary Fig. 25a) (as also reported in the original manuscript). We then examined the data matrix, aiming to dissect signals that can explicitly support 'magnoliids sister to monocots'. To this end, we found a clustering of 1,107 clusters (Fig. 6a). These genes are a collection of anchor pairs that potentially contributed to the grouping of magnoliids and monocots. Removing this part of the data changed the position of magnoliids (Supplementary Fig. 30). We checked the actual distribution of these particular anchor pairs (Supplementary Table 3) and found that some of them can form a signature block (example in Fig. 6b), which could be a shared derived character joining magnoliids and monocots, although we agree the alternative may still be possible. Admittedly, based on our current analysis, we cannot explicitly resolve the position of magnoliids, although we do think it is worthwhile to present our findings.

4/ The discussion about the brassicaceae phylogeny is also very interesting but would deserve some clarification. First it is written that according to reference 56, the "true" tree would be different from the tree supported by a majority of genes. It would be interesting to know how the true tree is known if the genes support another one (I suppose it is in the reference 56 but the argument misses in the present discussion), and to know what the true tree represents if genes follow another scenario. The authors are precautionous, they put quotes when using "true" and discuss a bit this notion, but it would be interesting to discuss this not from the point of view of the truth, but from the point of view of the processes. If a syntenic tree is closer to the species tree, does it mean that synteny is harder to introgress than single genes? Can we deduce something on the hybridization or introgression processes? Can synteny be viewed as a good alternative to sequences for phylogenetics when the sequence signal is blurred by transfers or introgressions?

Response: We have performed additional analyses in response to the reviewer's in-depth questions about why synteny can possibly reflect the true species tree compared to sequence alignment-based phylogenies in the presence of introgression. For more information and details, we refer to the main text, as well as to one of our earlier responses to reviewer 1.

And finally, minor remarks and probable typos:

- there is a "may related" which I would simply transform to "related"

Response: Changed.

- in Supp Fig 8, it would be useful to write which tree corresponds to which method

Response: We have revised the Figure as suggested, and also labeled the other trees for convenience (Supplementary Figs. 15-20).

- In the article it is written that a tutorial for the method is available at <https://github.com/zhaotao1987/SynNet-Pipeline> but I could not find it, the link leads to the code.

Response: We have updated the tutorial in wiki of this Github repository <https://github.com/zhaotao1987/SynNet-Pipeline/wiki>.

- "Syntenic genes" is not explained enough on line 502. Does it mean "a link between two homologous genes when they appear in a pairwise synteny block"? This is how I interpreted it but I did not see it explicated and it is crucial to understand the method.

Response: Yes, the reviewer's understanding is correct, we have clarified it in the revised version.

Reviewer #3 (Remarks to the Author):

This study uses synteny information to produce a phylogenetic tree that is largely congruent with the APG classification, which is almost entirely based on plastid information. Clearly the novel method retrieves results that are reliable, if judged from the perspective of congruence with other results. There are differences, but it is difficult to determine if this is a methodological problem or if the other data and methods used gave unreliable results. My opinion is that this paper tries to do too much in a format that does not allow a thorough examination of these novel methods. I suggest submission to journals where these methodological matters can be more thoroughly evaluated before launching into results, especially those that differ from previous studies.

Response: We appreciate the reviewer's comments with respect to the methodology. In the revised version of our manuscript, we have included work on simulations and benchmark data sets to strengthen the validity of our method and approach. As can be seen in the revised manuscript, the additional analyses indicate that, overall, our method shows good accuracy and is widely applicable (see also comments to other reviewer 1). We discuss how synteny-based phylogenetic inference can

complement traditional methods and could provide insights to long-standing controversial phylogenetic relationships.

I would also prefer to have results compared to those from datasets composed of nuclear gene data rather than just the APG (largely plastid) results. The differences could be just plastid versus nuclear trees.

Response: We actually had compared our synteny tree with a number of sequence alignment-based (SA) trees of nuclear genes (discussed in the original manuscript). We have reconstructed six SA trees on two sets of nuclear markers (BUSCO and CSSC) using three approaches (supermatrix, supertree, and multispecies coalescent) (Supplementary Figs 15-20). Our previous Figure 3 is a representative SA tree based on the concatenation of BUSCO genes. Most comparisons and discussions mentioned in the article refer to this tree (based on nuclear genes) and the synteny tree. Moreover, besides APG IV, we also have compared our synteny tree with the 1KP tree (Supplementary Fig. 24), which is based on large-scale transcriptome data (Leebens-Mack et al. 2019).

Reference

Leebens-Mack, J.H. et al. One thousand plant transcriptomes and the phylogenomics of green plants. *Nature* 574, 679-685 (2019)

Fundamentally, I wonder if it is strictly necessary to involve such complexity in methods to obtain a phylogenetic tree when other datasets/tree building methods have already been found to perform well. Given that the results are largely congruent with the established ideas of general angiosperm relationships and the differences focus mostly on results that have had a history of being problematic, this paper does not add a great deal to the literature about phylogenetic results. Whether the methods are reliable or problematic in some respects needs a lot more attention before proclaiming it to be useful.

Response: First, we have now added additional results on simulations and empirical data sets to validate our approach in the revised manuscript. We have demonstrated that the method performs well on many data sets, and that where it does not this is largely due to issues in synteny inference. We believe this points to an interesting issue (i.e., the reliability of synteny inference and the different performances of widely used synteny inference algorithms on taxonomically diverse data sets), which we believe however to be out of scope for our present study. For the reviewer's question about the necessity of the approach: as we have entered the post-genomic era, reference genomes are being sequenced and published rapidly. We believe there is (great) value in studying the evolution of genome organization and to investigate how conservation and diversification of genomic structure contributed to the delimitation and evolution of lineages. Studies using methods and models based on sequence-alignments often have conflicts. For plants, several famous contentious relationships are unsolved, and our analysis based on genomic structures (synteny) provides alternative evidence and a new/different perspective for plant systematics. The largely congruent result with traditional methods

for uncontroversial relationships confirmed the validity of the approach and highlighted the differences we observed. Thus, it can further be applied to lesser-known systems. Moreover, we believe the synteny network data structure we advocate here may provide a promising avenue for further research into model-based statistical phylogenetic inference from whole-genome data. In sum, phylogenetic inferences are troubled by a variety of factors, preventing us from achieving a fly-on-the-wall view of evolution. Clearly, different ways to look at and analyze what we have at our disposal, i.e., extant organisms, will be key to obtain a detailed view on the tree of life. We believe our synteny network based approach provides a promising avenue towards using whole-genome data for this endeavor.

Reviewers' Comments:

Reviewer #2:

Remarks to the Author:

The manuscript has been greatly improved compared to the previous version, which was already very interesting.

There is now a better description of the methods, some tests on simulated data and on several known clades, to evaluate it.

The discussion has been completed on all the difficult cases, and the merits of sequence phylogeny and synteny phylogeny are compared.

The bibliography is more complete.

All reviewers' comments have been addressed in depth, and a significant additional amount of work has been performed to examine every single suggestion.

The revision has turned an innovative method into a multidimensional phylogenetic study.

So I support publication of this version.